# Continuous Space-Time Video Super-Resolution via Event Camera

## Abstract

Continuous space-time video super-resolution (C-STVSR) aims to simultaneously enhance video resolution and frame rate at an arbitrary scale. Recently, implicit neural representation (INR) has been applied to video restoration, representing videos as implicit fields that can be decoded at an arbitrary scale. However, the highly ill-posed nature of C-STVSR limits the effectiveness of current INR-based methods: they assume linear motion between frames and use interpolation or feature warping to generate features at arbitrary spatiotemporal positions with **two** consecutive frames. This restrains C-STVSR from capturing rapid and **nonlinear motion** and **long-term dependencies** (*involving more than two frames*) in complex dynamic scenes. In this paper, we propose a novel C-STVSR framework, which captures both **h**olistic dependencies and **r**egional motions based on INR. It is assisted by an event camera – a novel sensor renowned for its high temporal resolution and low latency. To fully utilize the rich temporal information from events, we design a feature extraction consisting of (1) a regional event feature extractor – taking events as inputs via the proposed event temporal pyramid representation to capture the regional nonlinear motion and (2) a holistic event-frame feature extractor for long-term dependence and continuity motion. We then propose a novel INR-based decoder with spatiotemporal embeddings to capture long-term dependencies with a larger temporal perception field. We validate the effectiveness and generalization of our method on four datasets (both simulated and real data), showing the superiority of our method.

## 1 Introduction

The real world's visual information, *e.g.*, edge and object motion, is continuous, spanning both time and space dimensions. However, the limited I/O bandwidth and sensor size of modern systems (Delbracio et al., 2021; Parker, 2010) confines us to record videos at low frame rates and fixed resolutions. This limitation has profound repercussions across various computer vision applications, *e.g.*, encompassing immersive experiences in virtual reality (Zhang, 2020; Lee et al., 2020), traffic analysis in autonomous driving (Zou et al., 2023; Zhao et al., 2019). To address this limitation, recent research works (Chen et al., 2022; 2023b) have explored restoring videos with continuous resolutions and frame rates, referred to as Continuous Space-Time Video Super-Resolution (C-STVSR).

Recently, implicit neural representation (INR) has been applied to video restoration: it represents images or videos as neural fields that can be decoded at any resolution with a pointwise MLP decoder (Cao et al., 2023; Chen et al., 2023a). One of the seminal INR approaches is LIIF (Chen et al., 2021), which is designed for arbitrary-scale image SR. This line of research soon extended to the video domain. In this context of C-STVSR, VideoINR (Chen et al., 2022) employs a fixed STVSR model (Xiang et al., 2020) that extracts features from **two** consecutive video frames. Then, it introduces a temporal INR to generate inverse backward warping optical flow (Niklaus & Liu, 2020) to warp features. Lastly, it employs a spatial INR, similar to LIIF, to decode the RGB frame with arbitrary resolution. Building upon this, MoTIF (Chen et al., 2023b) improves VideoINR by using forward motion estimation, reducing gaps and holes in the temporal INR, which are typically caused by the randomness and discontinuities associated with backward warping (Park et al., 2021). *These methods depend solely on **two** successive RGB frames, rendering the task of predicting inter-frame motions ill-posed. Consequently,, it becomes challenging to accurately capture **highly dynamic motion** (e.g., regional high-speed or nonlinear movements) and to model **long-term dependencies** that extend across more than four frames.*

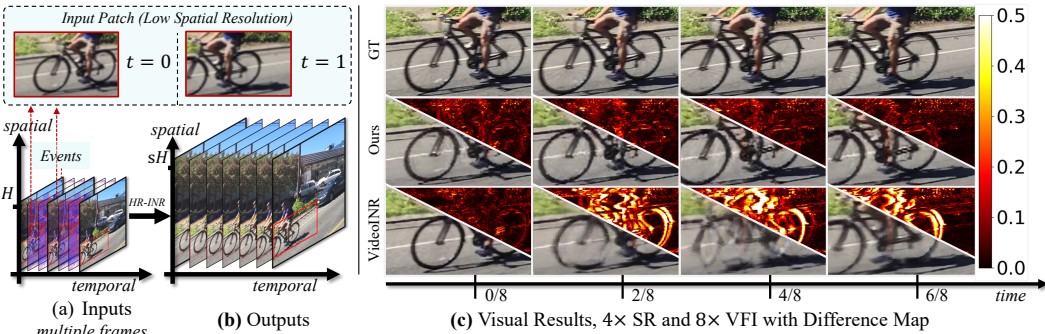

(a) Inputs
*multiple frames*

(b) Outputs

(c) Visual Results, 4× SR and 8× VFI with Difference Map

Figure 1: With event data as guidance, our method (HR-INR) takes in videos with low frame rates and resolution **(a)** and produces continuous space-time videos with arbitrary frame rate and resolution **(b)**. As shown in **(c)**, our method is able to recover the rotation of the bicycle wheels, which is unachievable by the prior method VideoINR (Chen et al., 2022).

**Motivation and Contributions.** Event cameras are bio-inspired sensors, known for their high temporal resolution and low latency ($< 1us$) (Zheng et al., 2023; Gallego et al., 2020; Wang et al., 2020a). Recent research has demonstrated the potential of events in guiding various video super-resolution (VSR) (Lu et al., 2023; Jing et al., 2021) and video frame interpolation (VFI) tasks (Tulyakov et al., 2021; 2022; He et al., 2022; Yu et al., 2021). *However, utilizing events to facilitate joint video super-resolution and frame interpolation is a challenging area yet to be explored.*

This paper introduces **HR-INR**, a novel INR-based method that leverages events for jointly guiding VSR and VFI. It adeptly captures regional, rapid motion and holistic, long-range motion dependencies, as shown in Fig. 1. To capture the regional motion, we propose Temporal Pyramid Representation (TPR) to construct a time-series pyramid structure around the pivotal timestamp of events (Sec. 3.1). Different from the evenly divided timeline representations, like voxel grids (Tulyakov et al., 2021; 2022; Gallego et al., 2020), time surfaces (Sironi et al., 2018), time moments (Han et al., 2021; Lu et al., 2023) and symmetric cumulative (Sun et al., 2022), TPR offers finer temporal granularity with less complexity and effectively captures rapid motion changes, as shown in Fig. 2 (a). Additionally, our method is capable of processing multiple frames and their associated events, which empowers it to estimate holistic, long-range motion that involves more than two frames.

Accordingly, we design two specialized feature extractors: the **r**egional event feature **e**xtractor (RE) and the **h**olistic event-frame feature **e**xtractor (HE), see Sec. 3.2. Both extractors are grounded in the Swin-Transformer architecture (Liang et al., 2021; Liu et al., 2022a; Liang et al.), renowned for its efficacy and efficiency in video enhancement tasks. RE is a lightweight network designed specifically for extracting local information from our event TPR. Meanwhile, HE employs a more sophisticated approach, utilizing long-term and multi-scale fusion strategies to integrate both events and frames. Consequently, our training and inference strategy requires HE to be invoked only once for multi-frame interpolation. Subsequently, the extracted features from RE and HE are fused as the output of the feature extraction module.

After fusing the regional and holistic features, we propose a novel INR-based spatial-temporal decoding module (Sec. 3.3). Our motivation is to avoid gaps and holes typically found in optical flow-based warping and multi-frame fusion, as identified in previous research (Chen et al., 2023b; 2022; He et al., 2022; Tulyakov et al., 2022). To accomplish this, we propose an implicit temporal embedding designed to transform timestamps into focused attention vectors on long-term features. This approach ensures attention is also given to long-distance dependencies, which is crucial for effectively modeling long-term temporal dependencies. Subsequently, inspired by LIIF (Chen et al., 2021), we employ spatial embedding to achieve arbitrary up-sampling in the spatial dimension.

We conducted experiments on two simulated and two real-world datasets. The results validate the superiority of our method and its excellent generalization capabilities on real-world datasets. *Our approach is the **first** event-based method to achieve continuous space-time video super-resolution, surpassing frame-based methods, as shown in Fig. 1. It also **excels** in individual VSR and VFI metrics compared to previous event-based methods.*

## 2 RELATED WORKS

**Space-time Video Super-Resolution** aims to enhance the resolution and frame rate of a video simultaneously (Haris et al., 2020; Kim et al., 2020; Xiang et al., 2020; Xu et al., 2021). In comparison to two-stage solutions, where VFI (Jiang et al., 2018; Xue et al., 2019; Niklaus & Liu, 2020; Niklaus et al., 2017; Cheng & Chen, 2020) and VSR (Liu et al., 2018; Yang et al., 2021; Yue et al., 2022; Wang et al., 2021; Tian et al., 2020; Isobe et al., 2020) methods are applied sequentially, simultaneous space-time video super-resolution reduces cumulative errors and leverages the natural relations between VFI and VSR methods. Zooming Slow-Mo (Xiang et al., 2020) uses temporal interpolation to generate missing frames and aligns temporal information using a deformable ConvLSTM network. Similarly, TMNet (Xu et al., 2021) extracts short-term and long-term motion cues in videos by modulating convolution kernels. *However, these methods cannot simultaneously achieve spatiotemporal resolution across **arbitrary scales**.*

**INR for VFI and VSR** have achieved space-time video super-resolution with arbitrary resolutions (Chen et al., 2022; 2023b) by learning videos implicit neural representations (INRs). These methods primarily estimate optical flows from **two** consecutive frames to warp features into arbitrary space-time coordinates, which are then decoded using MLP layers. *However, by relying on just two consecutive frames, these methods cannot inherently model long-term motions (involving three or more frames) and fail to accurately capture local, inter-frame, non-linear motions due to missing inter-frame information.*

**Event-guided VFI and VSR** seek to boost performance by incorporating the biologically inspired event cameras (Zheng et al., 2023). Previous works have demonstrated the potential of event-guided VFI, which mainly focus on modeling non-linear motion with events (Paikin et al., 2021; Tulyakov et al., 2021; Wu et al., 2022; Tulyakov et al., 2022; He et al., 2022; Song et al., 2022; 2023). EFI (Paikin et al., 2021) exclusively adopts the synthesis approach for intermediate frame generation. TimeLens (Tulyakov et al., 2021) and TimeLens++ (Tulyakov et al., 2022) employ events to model nonlinear motion correlations, integrating both synthesis and warping-centric approaches. Building on these advancements, CBM-Net (Kim et al., 2023), introduces a motion field to handle complex movements. *However, while these VFI methods utilize events to capture inter-frame motion, they fail to establish long-term dependencies beyond two frames and support simultaneous VSR.* The realm of event-guided video super-resolution has also been explored. E-VSR (Jing et al., 2021) highlights that high-frequency temporal information from events is beneficial to recovering high-frequency spatial information. Like our work, EG-VSR (Lu et al., 2023) employs events to comprehend INR, allowing for video ups-sampling with arbitrary scale. *However, the INR of EG-VSR cannot interpolate frames.* Contrasting these methods, we pioneer using events to enable concurrent VSR and VFI across arbitrary spatial-temporal scales, *i.e.,*, C-STVSR.

**Video Long-term Dependence Modeling** is a crucial aspect of VSR and VFI. For instance, BasicVSR (Chan et al., 2021) and BasicVSR++(Chan et al., 2022) enhance VSR performance by processing multi-frame inputs through an alignment module to model long-term motion correlations. Similarly, in the VFI domain, many methods (Suzuki & Ikehara, 2020; Nah et al., 2019; Zhang et al., 2020) employ RNN or LSTM to model sequences of frames, capturing long-term dependencies effectively. Furthermore, Zooming Slow-Mo (Xiang et al., 2020), TMNet (Xu et al., 2021), and RSST (Liang et al.) leverage multi-frame inputs in the joint task of VSR and VFI, showcasing the importance of integrating multiple frames for improved modeling of video dynamics. *However, current C-STVSR methods (Chen et al., 2022; 2023b), and event-based VFI methods (Tulyakov et al., 2021; He et al., 2022; Tulyakov et al., 2022; Kim et al., 2023), primarily rely on estimating optical flow between two consecutive frames. Therefore, they are challenging to handle multi-frames as inputs, inherently undermining their capability to model long-term dependencies.*

## 3 PROPOSED FRAMEWORK

Our proposed HR-INR framework is depicted in Fig. 2. The inputs of this framework are RGB frames $\boldsymbol{I}_{in} = \{I_{in}\}_i^{N_{in}} \in R^{N_{in} \times H \times W \times 3}$ and associated events $\boldsymbol{E}$. $H$ and $W$ denote the spatial resolution of frames and events. 3 means three channels of RGB. $N_{in}$ denotes the input number of frames. Furthermore, the framework outputs a video with an arbitrary frame rate and spatial resolution. In particular, we consider the output video as $\boldsymbol{I}_{out} = \{I_{out}\}_i^{N_{out}}$, consists of $N_{out}$

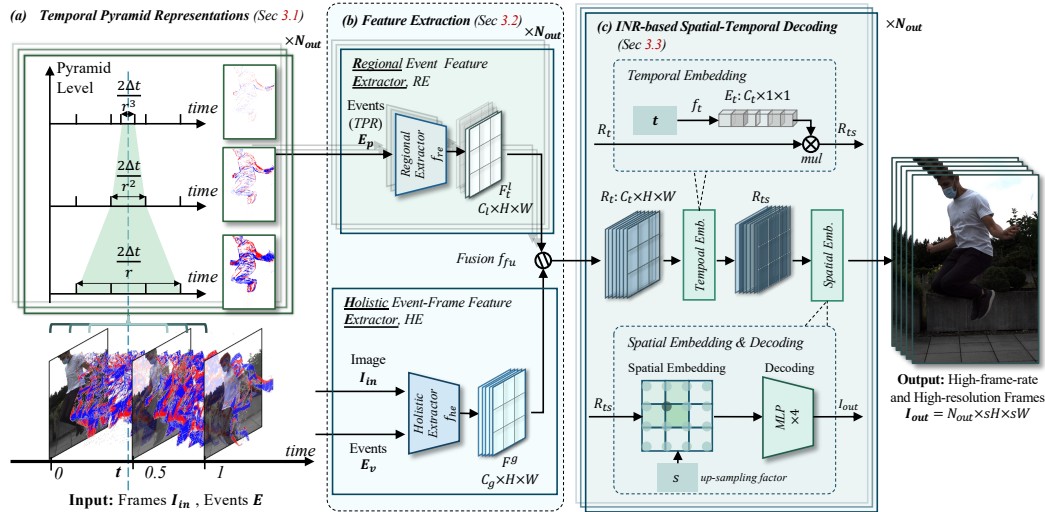

Figure 2: **Overview of our framework**. The inputs are multi-frame images and their corresponding events. The output is a video with enhanced frame rates and resolutions. Firstly, events proximate to a particular time point are transformed into Temporal Pyramid Representations (TPR) to capture motion at a more granular temporal level *(a)*. Secondly, TPRs, the comprehensive set of multi-frames and events, are directed into the feature extraction part *(b)*. Within this part, the Regional Events Feature Extractor and the Holistic Events Feature Extractor process the input separately. Lastly, the resulting features are then fused and inputted into an INR-based spatiotemporal decoding part *(c)*. Within this part, a temporal embedding is executed to capture features at a specific timestamp $t$, followed by spatial embedding with an up-sampling factor $s$ and decoding, culminating in the generation of frames at the desired resolution.

frames, each with a resolution of $(s \times H) \times (s \times W)$, where $s$ represents the up-sampling scale greater than 1. For the output $N_{out}$ frames, we denote the time corresponding to each frame as $\boldsymbol{T} = \{t\}_i^{N_{out}}$. For convenience, we also record the up-sampling scale $s$ and the time $\boldsymbol{T}$ as a part of inputs. Therefore, the mapping function $f_{hr}(.)$ of C-STVSR can be described by Eq. 1.

$$\boldsymbol{I}_{out} = f_{hr}\left(\boldsymbol{I}_{in}, \boldsymbol{E}, s, \boldsymbol{T}\right) \tag{1}$$

Our framework comprises three main components: First, Sec. 3.1 presents the event temporal pyramid representation (TPR), capturing regional dynamic motion and edges. Second, Sec. 3.2 elaborates on the feature extraction process using regional and holistic feature extractors. Third, Sec. 3.3 describes the INR-based spatiotemporal decoding.

**Input Frames and Events:** Our input frames, $\boldsymbol{I}_{in}$, consist of multiple frames with timestamps normalized to the $[0, 1]$ interval. We consider the events, $\boldsymbol{E}$, occurring within this time range. Each event point can be represented by $(x, y, t, p)$, signifying a change in pixel intensity at coordinates $(x, y)$ at time $t$; here, $p$ is $+1$ for increased brightness and $-1$ for a decrease. Benefiting from the event camera's high temporal resolution and low latency ($< 1us$), these event points are effectively considered continuous along the timeline. For a detailed exposition of the principles of event generation, please refer to the *Suppl. Mat.*.

### 3.1 TEMPORAL PYRAMID REPRESENTATION

Firstly, we represent the event stream $\boldsymbol{E}$ into a frame-like form that the network can process. To capture holistic motion, we partition all events during $[0, 1]$ of the timeline into $M$ equal intervals using a voxel grid (Tulyakov et al., 2021; Gallego et al., 2020), denotes as $\boldsymbol{E}_v$ with dimensions $M \times H \times W$. In practice, event representation methods like voxel grid (Tulyakov et al., 2021; Gallego et al., 2020) and its extended structure, symmetric cumulative event representation (Sun et al., 2022), achieve time granularity by uniformly dividing time into intervals with resolutions of $1/M$.

However, for C-STVSR, capturing intricate motion and edges requires a finer granularity. Merely increasing $M$ to enhance detail sharply raises computational costs; for instance, capturing $1/1000$ second intervals within a second necessitates expanding $M$ to 1000, which is computationally im-

practical. To address this, we introduce Temporal Pyramid Representation (TPR), leveraging the high temporal resolution of events while reducing representation complexity.

**Our idea:** The core of TPR is constructing a temporal pyramid where each successive layer's duration is $1/r, (r > 1)$ of the preceding one, leading to exponentially finer time granularity with additional layers. For instance, as illustrated in Fig. 2 (a), around any given time $t$, we define a surrounding time window $\Delta t$ and select an attenuation factor $r$. At the pyramid's $L$-th level, the events are within the time span of $[t - \Delta t/r^L, t + \Delta t/r^L]$. Each layer is further segmented into $M_p$ intervals, represented using a voxel grid. Accordingly, for an $L$-th layer, each layer contains $M_p$ moments within the TPR, and its finest time granularity, denoted by $\delta_t$, is as delineated in the Eq. 2:

$$\delta_t = \frac{2 \times \Delta t}{M_p \times r^L} \tag{2}$$

Therefore, for any time $t$, we construct the corresponding TPR $E_p$ with shape $L \times M_p \times H \times W$. We record the TPRs at all target timestamp as $\boldsymbol{E_p} = \{E_p\}_i^{N_{out}}$.

**Discussion:** The time granularity $\delta_t$ of TPR exponentially improves with the increase in layers, $L$. For an attenuation factor of $r = 3$ and a goal to detect motions as brief as $1/1000$ of a second within a $1s$ window ($2 \times \Delta t = 1$), we require only 7 TPR layers with each layer divided into 2 intervals ($M_p = 2$). Consequently, a TPR with dimensions $7 \times 2 \times H \times W$ suffices to discern motion down to $1/1000s$. Based on the above representation, we obtained $\boldsymbol{E_v}$, encapsulating holistic motion, and $\boldsymbol{E_p}$, which focuses on regional edges and motion.

### 3.2 HOLISTIC-REGIONAL FEATURE EXTRACTION

This module aims to extract features from regional TPRs $\boldsymbol{E_p}$, and the frame $\boldsymbol{I_{in}}$ and the holistic events $\boldsymbol{E_v}$ for INR-based spatial-temporal decoding. Accordingly, we introduce: **(1)** the regional event feature extractor, $f_{re}$, for dynamic motion and edges detail capture. **(2)** the holistic event-frame feature extractor, $f_{he}$, for long-term motion dependencies modeling across time and space.

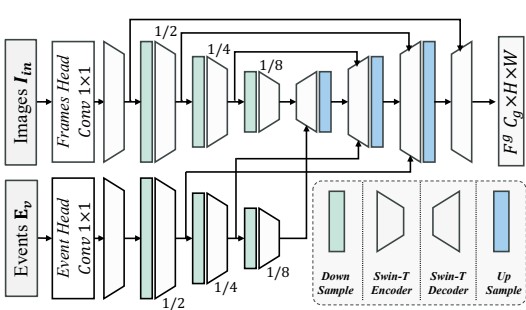

Figure 3: Holistic event-frame feature extractor. The down-sample module will halve the resolution. The up-sample module will double the resolution. The encoder and decoder have the same structure as Swin-Transformer (Liu et al., 2022b; 2021; Geng et al., 2022).

**Regional Event Feature Extraction:** The input of $f_{re}$ is TPR $E_p \in R^{L \times M_p \times H \times W}$ in each timestamp $t$. Given that $f_{re}$ is invoked $N_{out}$ times, its design needs to strike a balance between efficiency and the ability to model inter-level relationships to ensure that it can capture precise regional motion and edge details effectively. Initially, our method applies a convolution layer to increase the dimensions of the feature. Then, we use four Swin Transformer Encoder Blocks (STEB) (Liu et al., 2022b; 2021; Geng et al., 2022) to model the relationship between different pyramid levels. The STEB enjoys a large view field and a multi-head attention mechanism that proficiently models distant dependencies and conveys edge information across various pyramid levels, effectively capturing short-term motion. Notably, STEB is optimized with fewer parameters, boosting computational efficiency.

**Holistic Event-Frame Feature Extraction:** The inputs of $f_{he}$ is $\boldsymbol{I_{in}}$ and $\boldsymbol{E_v}$, as shown in Fig. 3. First, a convolutional layer processes both frames and events to increase dimensions. Motivated by the events feature manifest between successive frames to provide inter-frame motion information. We adjusted the first dimension of the event feature to $N_{in} - 1$ to account for inter-frame motion losses. We then incorporate the STEB to facilitate interactions across varied levels and spatial domains. To minimize computational overhead and expand the receptive field, we integrated a down-sampling module between STEBs, forming a multi-scale encoder. Each down-sampling iteration halves the resolution while maintaining the channel dimensions. After three iterations, the

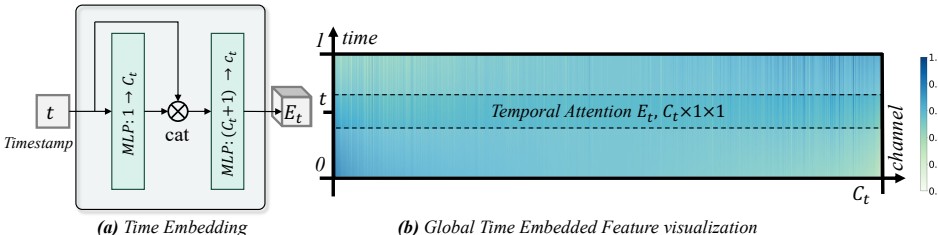

*(a) Time Embedding*          *(b) Global Time Embedded Feature visualization*

Figure 4: Temporal embedding. Given the input time $t \in [0, 1]$, the output is the temporal attention $E_t$ derived from a two-layer MLP. **(b)** presents a visualization of the trained $E_t$ during $[0, 1]$ on real-world dataset (Tulyakov et al., 2022).

feature resolution reduces to $1/8$ of its initial size, enlarging the receptive field. We then employ a Swin Transformer Decoder Block (Liu et al., 2022b; 2021; Geng et al., 2022) to fuse features at matching resolutions. Each of the first three STEBs is followed by an upsampling process, which doubles the resolution while maintaining the channel count. Ultimately, this process outputs the feature $F_g$. *For more details, see Suppl. Mat..*

Notably, to output $N_{out}$ frames, the holistic event-frame feature extractor $f_{he}$ is called once, capturing the comprehensive feature $F^g$. Subsequently, for each time $t$, $f_{re}$ extracts regional features $F_t^l$ from each TPR $E_p \in \boldsymbol{E_p}$. For each regional feature $F_t^l$, we use addition and $Cov1 \times 1$ operation as fusion function $f_{fu}$ to fuse with holistic feature $F^g$ to obtain the output $R_t$. For each time $t$, the whole process can be described by Eq. 3, where $E_p \in \boldsymbol{E_p}$ is the TPR at the specific timestamp $t$.

$$R_t = f_{fu}\left(F^g, F_t^l\right); F^g = f_{he}\left(\boldsymbol{I_{in}}, \boldsymbol{E_v}\right); F_t^l = f_{re}\left(E_p\right) \tag{3}$$

### 3.3 INR-BASED SPATIAL-TEMPORAL DECODING

In this section, we employ INR-based spatial-temporal decoding to effectively retrieve RGB frames at any desired time and resolution. To achieve this, we leverage a temporal INR to generate features at any timestamp and a spatial INR to upscale the features to any spatial resolution.

**Temporal Embedding:** We utilize learned temporal embedding as attention vectors to aggregate the fused feature in the channel dimension in a time-specific manner. At a given timestamp $t$, we first use a two-layer MLP to increase its dimension to $C_t$, resulting in a temporal attention vector (as illustrated in Fig. 4 (a)). The visualization of the learned temporal attention is depicted in Fig. 4 (b), exhibiting variations across both time and channel dimensions. This attention vector is then multiplied directly with $R_t$ to generate the temporal embedded feature $R_{ts}$. This temporal INR allows for a larger temporal perception field without the need for estimating optical flows. Next, a $1 \times 1$ convolution is applied to compress $R_{ts}$ to $C_{ts}$ dimensions, reducing the complexity of the spatial embedding and decoding.

**Spatial Embedding and Decoding:** To upscale the temporal embedded features to any desired spatial scale, we utilize a similar approach to previous works (Chen et al., 2022; 2023b; 2021). We query the four nearest neighbors in the temporal embedded feature for each spatial coordinate and concatenate these with their distances for spatial embedding. A four-layer MLP decoder computes the RGB values, which are then aggregated through area-weighted interpolation for arbitrary-scale super-resolution. Similar to (Lu et al., 2023), we use the *Charbonnier loss* (Lai et al., 2018) as the fundamental loss function for training.

## 4 EXPERIMENTS

To facilitate a comprehensive comparison between the frame-base (Chen et al., 2022; 2023b) and event-guided methods (Lu et al., 2023; Tulyakov et al., 2021; 2022; Jing et al., 2021), we employed two simulated datasets (Su et al., 2017; Nah et al., 2017) and two real-world datasets (Tulyakov et al., 2022; Scheerlinck et al., 2019a) for our experiments **(1) Adobe240 dataset** (Su et al., 2017) includes 133 videos, each with $720 \times 1280$ resolution and a $240fps$ frame rate. Following established protocols (Chen et al., 2022;

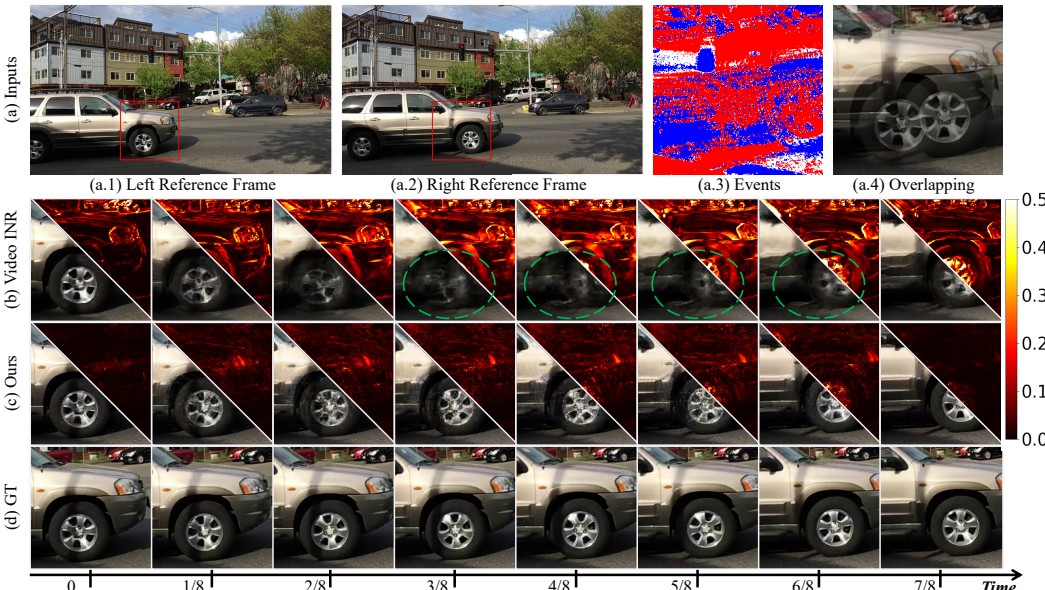

Figure 5: 7-*skip* frame interpolation visualization results in Adobe240 dataset (Su et al., 2017). Our method **(c)** more effectively captures the **rotating wheels** compared to the VideoINR **(b)**, which tends to show noticeable holes and gaps. Green circles highlight obvious holes and gaps.

2023b; Xu et al., 2021), the dataset is divided into 100 training, 16 validation, and 17 testing subsets. We employed the widely-used event simulation method vid2e (Gehrig et al., 2020), which accounts for real noise distribution, enhancing our model's robustness and generalization capabilities. **(2) GoPro dataset** (Nah et al., 2017) featuring the same resolution and frame-rate with Adobe240 dataset, comprises 22 training and 11 test videos. Given its compact size, previous studies (Chen et al., 2023b; 2022) have primarily employed the test set for quantitative analysis. We follow this to be consistent with established practice. For datasets **(1)** and **(2)** during 7-*skip* frame interpolation, we input 4 frames where each pair of adjacent frames is separated by 7 frames, allowing a total of 25 frames to serve as GTs. **(3) BS-ERGB** (Tulyakov et al., 2022) recorded using a spectroscope, includes real-world paired events and frames. Previous studies, *e.g.*, TimeLens++(Tulyakov et al., 2022), initially pre-trained on the Vime90k simulation dataset(Xue et al., 2019) before fine-tuning on BS-ERGB. In contrast, we opted to pre-train our model on the Adobe240 dataset before fine-tuning. Notably, the Vime90k dataset is larger in scale than the Adobe240 dataset. During fine-tuning, we also used perceptual loss (Johnson et al., 2016) with weight 0.1 to be consistent with previous methods (Tulyakov et al., 2021; 2022) for fair comparison. **(4) CED** (Scheerlinck et al., 2019a) is a real-world dataset in VSR. To fairly compare the previous research (Jing et al., 2021; Lu et al., 2023), only this data set is used for training for VSR comparison.

**Implementation Details:** Our model is trained using Pytorch (Paszke et al., 2019), employing the Adam optimizer (Kingma & Ba, 2014). Referring to the VideoINR (Chen et al., 2022), our training consists of two stages. **(1)** Train frame interpolation under a fixed spatial up-sampling ($4\times$), over 70 epochs, starting with a learning rate of $5e - 4$. **(2)** Train frame interpolation under random space upsampling rate in $\mathcal{U}(1, 8)$, spanning 30 epochs with the learning rate $5e - 5$. We randomly choose 20 frames from a pool of 25 frames as ground truth. Data augmentation is implemented via *Random Crop*, extracting $512 \times 512$ areas from frames and events, with the input resolution dynamically determined by a randomly chosen upsampling ratio $s$. To optimize memory usage and accelerate speed, we implemented the mixed precision strategy (Micikevicius et al., 2017; Das et al., 2018). All experiments are performed on an NVIDIA A800 computing card. *Please refer to the Suppl. Mat. for more details.*

**Evaluation:** To ensure the fairness, we adopted PSNR (Zhang et al., 2018), SSIM (Wang et al., 2004), and LPIPS (Zhang et al., 2018) as quantitative evaluation metrics. Aligning with prior works (Chen et al., 2022; 2023b) for consistency, we use only the $Y$-channel for GoPro and Adobe datasets and all three RGB channels for BS-ERGB and CED datasets.

Table 1: Quantitative metrics (PSNR/SSIM) with 7-*skip* VFI and $4\times$ VSR. *Center Average* remain consistent with previous work (Chen et al., 2022; 2023b).

| VFI | VSR | Params ($M$) | GoPro-*Center* | GoPro-*Average* | Adobe-*Center* | Adobe-*Average* |
|---|---|---|---|---|---|---|
| | Bicubic | 19.8 | 27.04/0.7937 | 26.06/0.7720 | 26.09/0.7435 | 25.29/0.7279 |
| Su-SloMo | EDVR | 19.8+20.7 | 28.24/0.8322 | 26.30/0.7960 | 27.25/0.7972 | 25.95/0.7682 |
| | BasicVSR | 19.8+6.3 | 28.23/0.8308 | 26.36/0.7977 | 27.28/0.7961 | 25.94/0.7679 |
| | Bicubic | 29.2 | 26.50/0.7791 | 25.41/0.7554 | 25.57/0.7324 | 24.72/0.7114 |
| QVI | EDVR | 29.2+20.7 | 27.43/0.8081 | 25.55/0.7739 | 26.40/0.7692 | 25.09/0.7406 |
| | BasicVSR | 29.2+6.3 | 27.44/0.8070 | 26.27/0.7955 | 26.43/0.7682 | 25.20/0.7421 |
| | Bicubic | 24.0 | 26.92/0.7911 | 26.11/0.7740 | 26.01/0.7461 | 25.40/0.7321 |
| DAIN | EDVR | 24.0+20.7 | 28.01/0.8239 | 26.37/0.7964 | 27.06/0.7895 | 26.01/0.7703 |
| | BasicVSR | 24.0+6.3 | 28.00/0.8227 | 26.46/0.7966 | 27.07/0.7890 | 26.23/0.7725 |
| TimeLens | EG-VSR | 72.2+2.45 | 28.85/0.8678 | 27.54/0.8293 | 28.11/0.8441 | 27.42/0.8269 |
| CBMNet | EG-VSR | 22.2+2.45 | 29.22/0.8686 | 28.51/0.8493 | 28.28/0.8553 | 27.89/0.8334 |
| Zooming Slow Mo | | 11.10 | 30.69/0.8847 | -/- | 30.26/0.8821 | -/- |
| TMNet | | 12.26 | 30.14/0.8692 | 28.83/0.8514 | 29.41/0.8524 | 28.30/0.8354 |
| Video INR-*fixed* | | 11.31 | 30.73/0.8850 | -/- | 30.21/0.8805 | -/- |
| Video INR | | 11.31 | 30.26/0.8792 | 29.41/0.8669 | 29.92/0.8746 | 29.27/0.8651 |
| MoTIF | | 12.55 | 31.04/0.8877 | 30.04/0.8773 | 30.63/0.8839 | 29.82/0.8750 |
| HR-INR (Ours) | | 8.27 | **31.97/0.9298** | **32.13/0.9371** | **31.26/0.9246** | **31.11/0.9216** |

Table 2: More quantitative comparisons using PSNR/SSIM on the GoPro dataset. **Bold** indicates the best performance.

| Temporal Scale | Spatial Scale | Su-SloMo + LIIF | DAIN + LIIF | TMNet | Video INR | MoTIF | Ours |
|---|---|---|---|---|---|---|---|
| $12\times$ | $4\times$ | 25.07/0.7491 | 25.14/0.7497 | 26.38/0.7931 | 27.32/0.8141 | 27.77/0.8230 | **28.87/0.8854** |
| $12\times$ | $6\times$ | 22.91/0.6783 | 22.92/0.6785 | - | 24.68/0.7358 | 26.78/0.7908 | **27.14/0.8173** |
| $16\times$ | $4\times$ | 24.42/0.7296 | 24.20/0.7244 | 24.72/0.7526 | 25.81/0.7739 | 25.98/0.7758 | **27.29/0.8556** |
| $16\times$ | $6\times$ | 23.28/0.6883 | 22.80/0.6722 | - | 23.86/0.7123 | 25.34/0.7527 | **26.09/0.7954** |
| $6\times$ | $1\times$ | - | - | - | 32.34/0.9545 | 34.77/0.9696 | **38.53/0.9735** |
| $1\times$ | $4\times$ | - | - | 33.02/0.9206 | 32.26/0.9198 | **33.84**/0.9328 | 33.51/**0.9417** |

## 4.1 COMPARISON EXPERIMENTS

**Space-time Super-resolution:** We conduct space-time super-resolution comparison experiments on the Adobe240 and GoPro datasets. We categorized the comparison methods into three groups. **(I)** Frame-based cascade methods: VFI methods, *e.g.*, Super SloMo (Jiang et al., 2018) and DAIN (Bao et al., 2019) followed by VSR methods, *e.g.*, EDVR (Wang et al., 2019) and BasicVSR (Chan et al., 2021). **(II)** Fixed STVSR methods: *e.g.*, Zooming Slow-Mo (Xiang et al., 2020) and TMNet (Xu et al., 2021). **(III)** Frame-based C-STVSR methods: VideoINR (Chen et al., 2022) and MoTIF (Chen et al., 2023b). The numerical comparison is presented in Tab. 1. It is evident that C-STVSR methods consistently outperform cascade and fixed STVSR methods.Our method achieves the highest performance in both datasets with the smallest model size. In the GoPro dataset, our method improves the center frame by 0.93 $dB$ and 0.0415 SSIM, and on average by 2.09 $dB$ and 0.0598 SSIM compared to the best method, MoTIF (Chen et al., 2023b). Similarly, in the Adobe240 dataset, our method outperforms MoTIF (Chen et al., 2023b) by 0.63 $dB$ and 0.0407 SSIM for the -*center*, and on -*average* by 1.29 $dB$ and 0.0466 SSIM. It is worth noting that the difference between our method and other methods is more pronounced for the -*average* frames than the -*center* frame. This observation suggests that our method demonstrates enhanced **temporal stability** and superior adaptability across varying interpolation intervals, an advantage not shared by previous methods (Chen et al., 2022; 2023b). For more analysis, please refer to Sec. 4.2 and Fig. 12 in appendix.

Tab. 2 presents additional comparative experiments for arbitrary spatial and temporal super-resolution. Our method consistently outperforms other methods, even in extreme space-time up-sample scales, such as $16\times$ temporal upscale and $6\times$ spatial upscale. The visualization results, Fig. 1 (c) and Fig. 5, also demonstrate that our method effectively models regional nonlinear motion, *e.g.*, the wheel rotation — a capability not achieved by previous method VideoINR (Chen et al., 2022). *For more visualization results, please refer to the Suppl. Mat.*

Table 3: Temporal super-resolution results, *i.e.,*, VFI, on BS-ERGB dataset (Tulyakov et al., 2022).

| Methods | Params (M) | Event | 1-skip PSNR ↑ | SSIM ↑ | LPIPS ↓ | 3-skip PSNR ↑ | SSIM ↑ | LPIPS ↓ |
|---|---|---|---|---|---|---|---|---|
| FLAVR (Kalluri et al., 2023) | - | ✗ | 25.95 | - | 0.086 | 20.90 | - | 0.151 |
| DAIN (Bao et al., 2019) | 24.0 | ✗ | 25.20 | - | 0.067 | 21.40 | - | 0.113 |
| Super SloMo (Jiang et al., 2018) | 19.8 | ✗ | - | - | - | 22.48 | - | 0.115 |
| QVI (Xu et al., 2019) | 29.2 | ✗ | - | - | - | 23.20 | - | 0.110 |
| TimeLens (Tulyakov et al., 2021) | 72.2 | ✓ | 28.36 | - | 0.026 | 27.58 | - | 0.031 |
| TimeLens++ (Tulyakov et al., 2022) | 53.9 | ✓ | 28.56 | - | 0.022 | 27.63 | - | 0.026 |
| CBMNet (Kim et al., 2023) | 15.4 | ✓ | 29.32 | 0.815 | - | 28.46 | 0.806 | - |
| CBMNet-*Large* (Kim et al., 2023) | 22.2 | ✓ | 29.43 | 0.816 | - | 28.59 | 0.808 | - |
| HR-INR (Our) | 8.3 | ✓ | **29.66** | **0.828** | **0.011** | **28.59** | **0.814** | **0.021** |

Table 4: Spatial super-resolution results on CED (Scheerlinck et al., 2019b). * denotes values from pre-trained models.

| Methods | Params (M) | Events | 4× PSNR ↑ | SSIM ↑ | 2× PSNR ↑ | SSIM ↑ |
|---|---|---|---|---|---|---|
| RBPN (Haris et al., 2019) | 12.18 | ✗ | 29.80 | 0.8975 | 36.66 | 0.9754 |
| VideoINR* (Chen et al., 2022) | 11.31 | ✗ | 25.53 | 0.7871 | 26.77 | 0.7938 |
| E-VSR (Jing et al., 2021) | 412.42 | ✓ | 30.15 | 0.9052 | 37.32 | 0.9783 |
| EG-VSR (Lu et al., 2023) | 2.45 | ✓ | 31.12 | 0.9211 | 38.69 | 0.9771 |
| HR-INR (Our) | 8.27 | ✓ | **32.15** | **0.9658** | **42.01** | **0.9905** |

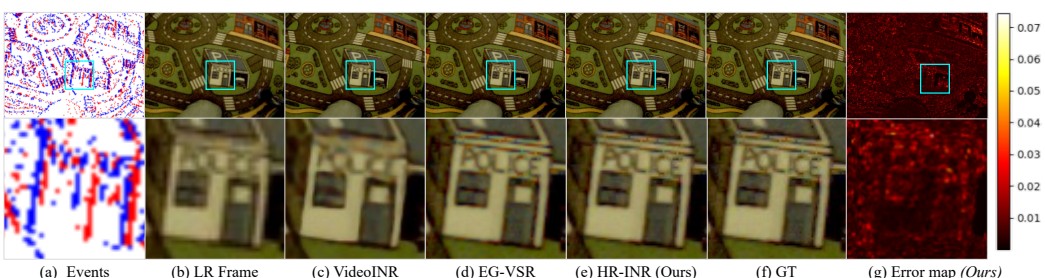

(a) Events  (b) LR Frame  (c) VideoINR  (d) EG-VSR  (e) HR-INR (Ours)  (f) GT  (g) Error map *(Ours)*

Figure 6: 4× video super-resolution visualization results in CED dataset (Scheerlinck et al., 2019b).

**Separate Comparison of Event-based VFI and VSR:** We also compared our method with previous approaches in separate VFI (Tulyakov et al., 2022; 2021; Kim et al., 2023) and VSR (Lu et al., 2023; Jing et al., 2021) tasks. The results can be seen in Tab. 3 and Tab. 4 respectively. In the VFI task, our method surpasses TimeLens++ (Tulyakov et al., 2022) by 1.1 $dB$ for 1-*skip* and 0.17 $dB$ for 3-*skip* VFI, and CBMNet (Kim et al., 2023) by 0.23 $dB$ for 1-*skip*. Additionally, our model's size is merely 1/7 and 1/3 that of TimeLens++ (Tulyakov et al., 2022) and CBMNet (Kim et al., 2023), respectively. In the visualization aspect, our approach excels in modeling non-linear motion and long-term dependencies, as evident in Fig. 13 in appendix. It precisely forecasts the positions of small balls (Fig. 13 (1.d)) and the soccer ball (Fig. 13 (2.d)) at intermediate timestamps, outperforming previous methods.

For the VSR task, our method outperforms EG-VSR (Lu et al., 2023) by 1.03 $dB$ and 0.447 SSIM for 4 × super-resolution, and by 3.02 $dB$ and 0.0134 SSIM for 2 × super-resolution. Fig. 6 demonstrates our model's effective VSR performance on the real-dataset CED, highlighting its ability to reproduce sharp edges and robustness to noise. *For more visualization results, see the Suppl. Mat.*

## 4.2  Ablation and Analytical Studies

Ablation and analytical studies conducted on the Adobe240 (Su et al., 2017) and BS-ERGB (Tulyakov et al., 2022) datasets unveiled several critical insights. On the Adobe240 dataset, we executed simultaneous 7-*skip* VFI and 4× VSR tests, as shown in Tab. 5 and Tab. 6, while on the BS-ERGB dataset, 1-*skip* and 3-*skip* VFI were performed in Tab. 7. **Events Gain**: Tab. 5-*Case#1* shows that with events input replaced as zero and unchanged network architecture, PSNR and SSIM significantly drop, highlighting the importance of events for temporal motion learning. Adding events alone substantially raised PSNR by $2.85dB$ and SSIM by $0.06$. **Event TPR Enhancement:** Incorporating event TPR enhances performance, with PSNR and SSIM increasing with layer count, peaking at

Table 5: Ablation studies in Adobe-*Average* Su et al. (2017) (4× and 7-*skip*). The † symbol marks the line for comparison with other lines.

| Case | Events | TPR | Temporal Embedding | Temporal Dim ($C_t$) | Input Frames | PSNR ↑ | SSIM ↑ |
|------|--------|-----|--------------------|----------------------|--------------|--------|--------|
| Case#1 | ✗ | ✗ | *Learning* | 640 | 4 | 26.84 ($-4.27$) | 0.8366 ($-0.0850$) |
| Case#2 | ✓ | ✗ | *Learning* | 640 | 4 | 29.69 ($-1.41$) | 0.8974 ($-0.0242$) |
| Case#3 † | ✓ | ✓ | *Learning* | 640 | 4 | **31.11** | **0.9216** |
| Case#4 | ✓ | ✓ | *Sinusoid* | 640 | 4 | 30.42 ($-0.69$) | 0.9120 ($-0.0096$) |
| Case#5 | ✓ | ✓ | *Learning* | 320 | 4 | 28.44 ($-2.67$) | 0.8700 ($-0.0516$) |
| Case#6 | ✓ | ✓ | *Learning* | 640 | 2 | 30.41 ($-0.70$) | 0.9151 ($-0.0065$) |
| Case#7 | ✓ | ✓ | *Learning* | 640 | 3 | 30.72 ($-0.39$) | 0.9174 ($-0.0042$)- |

Table 6: Ablation studies for TPR levels and moments in Adobe-*Average* Su et al. (2017) (4× and 7-*skip*). The † symbol marks the line for comparison with other lines. "Captured Moment" refers to the temporal resolution of the last layer of the TPR, which is calculated by Eq. 2.

| Case | TPR Level ($L$) | TPR Moments ($M_p$) | Captured Moment | PSNR ↑ | SSIM ↑ |
|------|-----------------|---------------------|-----------------|--------|--------|
| Case#1 | 3 | 3 | 1 / 81 | 29.93 ($-1.18$) | 0.9011 ($-0.0205$) |
| Case#2 | 5 | 3 | 1 / 729 | 30.32 ($-0.79$) | 0.9165 ($-0.0051$) |
| Case#3 | 7 | 3 | 1 / 6561 | 30.78 ($-0.33$) | 0.9187 ($-0.0029$) |
| Case#4 † | 7 | 9 | 1 / 19683 | *31.11* | *0.9216* |
| Case#5 | 7 | 18 | 1 / 39366 | **31.18** ($+0.07$) | **0.9228** ($+0.0012$) |

Table 7: Ablation in BS-ERGB (TimeLens++) dataset Tulyakov et al. (2022).

| TPR | 1-*skip* | | | 3-*skip* | | |
|-----|----------|--------|---------|----------|--------|---------|
| | PSNR ↑ | SSIM ↑ | LPIPS ↓ | PSNR ↑ | SSIM ↑ | LPIPS ↓ |
| ✗ | 28.25 | 0.8187 | 0.018 | 26.65 | 0.7867 | 0.039 |
| ✓ | **29.66** ($+1.41$) | **0.8281** ($+0.0094$) | **0.011** ($-0.007$) | **28.59** ($+1.94$) | **0.8140** ($+0.0273$) | **0.021** ($-0.018$) |

seven layers, as shown in Tab. 5-*Case#2* and Tab. 6. Specifically, in Tab. 5, as the TRP Level $L$ increases, the moments captured by the TPR become more precise. When the TRP Level rises from 3 to 7, the PSNR exhibits an increase of approximately 1.18. Furthermore, the model's performance also improves with the increase in TPR Moments $M_p$. However, during the phases where both $L$ and $M_p$ are relatively high, this improvement tends to plateau. The TPR also demonstrates enhancement on the BS-ERGB dataset, as shown in Tab. 7, yielding an increase of $0.92dB$ for 1-*skip* and an improvement of $1.13dB$ for 3-*skip*. This indicates that the benefits of TPR become more pronounced with the increase in the number of skips. **Time Embedding Method:** Table 5-*Case#3#4* shows Sinusoid embedding's results. It's outperformed by learning-based methods, confirmed by previous research (Ramasinghe & Lucey, 2023; Attal et al., 2022), due to their superior high-frequency information capture. **Temporal Dimension Impact:** The INR temporal dimension significantly influences performance. Lowering the dimensions from 640 to 320 degrades performance in Tab. 5-*Case#3#5*, suggesting a reduction in temporal detail capture. Conversely, expanding the dimension to 960 poses instability risks (*e.g.*, $NAN$ errors). This highlights the need to balance dimensionality and training stability. **Input Frames:** Tab. 5-*Case#3#6#7* illustrates the impact of varying input frame counts on the final results. We observed a performance decrease of 0.70 $dB$ and 0.39 $dB$ when inputting two and three frames, respectively, compared to four frames. This indicates a clear advantage of multi-frame inputs in modeling longer-term dependencies. Moreover, even with two frames, our method also outperforms previous works (Chen et al., 2022; 2023b).

## 5 CONCLUSION

Our work introduced the first event-guided continuous space-time video super-resolution method. The main contributions are: **(I)** Event temporal pyramid representation for capturing short-term dynamic motion; **(II)** A feature extraction process combining holistic and regional features to manage motion dependencies; **(III)** A spatiotemporal decoding process based on implicit neural representation, avoiding traditional optical flow and achieving stable frame interpolation through temporal-spatial embedding.

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

## APPENDIX OVERVIEW

The appendix of this paper consists of five sections:

1. **(Sec. A) Imaging Principle of Events and the Guidance of C-STVSR Task:**
   This section introduces the event generation model, providing insights into why events are effective for C-STVSR.

2. **(Sec. B) More Details about the Network Structure:**
   Detailed information on the network architecture is provided, including comprehensive descriptions of the inputs, outputs, and intermediate processes of each module.

3. **(Sec. C) More Details about the Experimental Settings:**
   This section describes additional experimental settings to ensure the accurate replication of our method.

4. **(Sec. D) Additional Experimental Results and Analysis:** Further experimental analysis and results are offered, demonstrating the superiority of our approach. The key subsections include:

   - **More Visualization with Compare Methods:** Visual comparisons highlighting the advantages of our method over competing approaches.
   - **Capturing Millisecond Motion with Event TPR:** Analysis of TPR's capability to capture fine-grained local motion.
   - **Analysis of Input Frames and Long-Distance Motion Modeling:** Study of the impact of multi-frame inputs on interpolation performance.
   - **More Metric Evaluation Results on Real-World Datasets:** Quantitative comparisons using no-reference metrics (NIQE and PI).
   - **More Experiments on the APLIX-VSR Dataset:** Extended evaluations on various upscaling factors.
   - **Stability of VFI in Various Timestamps:** Analysis of temporal coherence across different timestamps.
   - **Bad Case Analysis:** Identification and discussion of failure cases.
   - **Inference Time Analysis:** Evaluation of inference efficiency.

5. **(Sec. E) Additional Visualization Results:**
   This section presents more visual materials, including images and videos.

## A  IMAGING PRINCIPLE OF EVENTS AND THE GUIDANCE OF C-STVSR TASK

Events are discrete points capturing the positive and negative changes in pixel brightness. Their generation hinges on brightness alterations within the logarithmic domain. Specifically, an event point $e = (x, y, t, p)$ is triggered and logged upon meeting certain criteria. Suppose $L(x, y, t)$ represents the brightness at point $(x, y)$ at any given time $t$. The event is recorded if the absolute difference $\Delta L = log\left(L(x, y, t)\right) - L\left(x, y, t - \Delta t\right)$ surpasses a predetermined threshold $C$, as formulated as Eq. 4.

$$p = \begin{cases} +1, \Delta L > C \\ -1, \Delta L < -C \end{cases} \tag{4}$$

Utilizing Eq. 4, the processing pipeline for a specified pixel at coordinates $(x, y)$ at any given time $t$ and $t'$ can be delineated by Eq. 5.

$$L(x, y, t) = I(x, y, t') \times \exp(C \int_{t'}^{t} p \, dt) \tag{5}$$

Utilizing Eq. 5, coupled with corresponding events, the intensity frame at a given time enables the computation of intensity frames for alternate times, facilitating video frame interpolation. However, events typically contain noise, and employing Eq. 5 directly cannot produce optimal outcomes (Pan et al., 2019).

Numerous studies (Paikin et al., 2021; Tulyakov et al., 2021; Zhang & Yu, 2022) have demonstrated the effectiveness of employing neural networks to guide event-based modeling in robust frame interpolation. Additionally, the high temporal resolution of events aids in high spatial resolution conversion, a finding corroborated by prior research (Lu et al., 2023; Jing et al., 2021).

In conclusion, comprehending the event generation model reveals its substantial benefits for frame interpolation and VSR tasks, making it a natural guide for continuous space-time video super-resolution (C-STVSR) tasks.

## B  More Details about Network Structure

Owing to space constraints in the main text, this section provides a more detailed account of the network's results. The description begins with the Regional Event Feature Extraction component, covering its preprocessing steps and the architecture of the Swin Transformer Encoder Blocks (STEB). Subsequently, the Holistic Event-Frame Feature Extraction is detailed, which, besides STEB, incorporates the Swin Transformer Decoder Block (STDB). In summary, this section elaborates on the preprocessing for both feature extraction components and details the structures of STEB and STDB.

### B.1  Regional Event Feature Extraction:

This section delves into Regional Event Feature Extraction, denoted as $f_{re}$. $f_{re}$ receives Temporal Pyramid Representation (TPR) $E_p$ corresponding to timestamp $t$, formatted as $R^{L \times M_p \times H \times W}$. Here, $L$ signifies the number of TPR layers, while $M_p$ indicates the count of moments within each layer. $f_{re}$ involves two primary stages: preprocessing and feature extraction via STEB. Specifically, this preprocessing first increases its dimension to $C_r$ through a $1 \times 1$ convolution operation. This preprocessing alters the TPR's shape to $L \times C_r \times H \times W$. Feature extraction is then performed using STEB, where both the input and output retain their shape throughout the process.

**Swin Transformer Encoder Blocks (STEB):** The STEB structure, proven effective in video super-resolution and frame interpolation (Liu et al., 2022b; 2021; Geng et al., 2022), is adopted in our design. The input and output shapes of STEB are represented as $L \times C \times H \times W$ for clarity. Initially, for a window size of $M \times M$, specifically $4 \times 4$ in our implementation, the input is partitioned into disjoint windows of $(M \times M) \times N \times (H/M) \times (W/M) \times C$ dimensions. Then, each window is compressed to form a feature map of shape $(M \times M) \times (N \times H \times W/M^2) \times C$. Following this, Layer Normalization (Ba et al., 2016) and window-based multi-head self-attention (Liu et al., 2021; Geng et al., 2022) are computed for each window, succeeded by further transformation via another Layer Normalization and a Multi-Layer Perception. Shifted window-based multi-head self-attention (Liu et al., 2021; Geng et al., 2022) is then employed to establish cross-window connections. After applying one STEB structure, a second STEB is introduced with an identical configuration, except the input feature window is offset by $(M/2) \times (M/2)$. In total, four STEBs are utilized for comprehensive feature extraction in the Regional Event Feature Extraction.

### B.2  Holistic Event-Frame Feature Extraction:

This section details the Holistic Event-Frame Feature Extraction ($f_{he}$) module. The $f_{he}$ module processes inputs $\boldsymbol{v}I_{in}$ and $\boldsymbol{v}E_v$ to generate the output $F^g$. Input $\boldsymbol{v}I_{in}$ has a shape of $N \times 3 \times H \times W$, whereas $\boldsymbol{v}E_v$ is structured as $M \times H \times W$, with $M$ indicating the count of moments derived from the events. The output, $F^g$, has a structure of $C_g \times H \times W$. Initially, $f_{he}$ transforms $\boldsymbol{v}I_{in}$ and $\boldsymbol{v}E_v$. $\boldsymbol{v}I_{in}$ and $\boldsymbol{v}E_v$ are transformed into features with dimensions $N \times C \times H \times W$ and $N - 1 \times C \times H \times W$ using $1 \times 1$ convolution and reshaping, followed by processing through a STEB module. The process then employs a combination of downsampling and STEB to extract features across multiple scales, each possessing a distinct view field. Notably, STEB maintains the same resolution and channel count in both its input and output. Next, the Swin Transformer Decoder (STDB) is utilized for fusing and decoding the dual-modal features. Each STDB block receives three types of inputs: the

preceding STDB's output and the outputs from the $vI_{in}$ and $vE_v$ encoder sections, all at a matching resolution. In practice, the outputs from the $vI_{in}$ and $vE_v$ encoders are concatenated, transformed into queries via an MLP layer, and then fused utilizing a multi-head attention mechanism. Notably, the first STDB module lacks a preceding STDB output, hence defaulting this input to zero. The final output, $F^g$, is derived from the last STDB's output, post reshaping and $1 \times 1$ convolution. In summary, $f_{he}$ employs a sequence of intricate yet efficacious transformation and fusion procedures, designed to extract multifaceted features from inputs $vI_{in}$ and $vE_v$ for subsequent tasks in video super-resolution and frame interpolation.

## C  More Details about Experiments Setting

### C.1  More Details about Datasets

To facilitate a comprehensive comparison between the frame-base (Chen et al., 2022; 2023b) and event-guided methods (Lu et al., 2023; Tulyakov et al., 2021; 2022; Jing et al., 2021), we employed four datasets: two simulated (Adobe240 (Su et al., 2017) and GoPro (Nah et al., 2017)) and two real-world (HS-ERGB (Tulyakov et al., 2022) and CED (Scheerlinck et al., 2019a)) for our experiments.

**1) Adobe240 Dataset (Su et al., 2017):** This dataset comprises 133 videos, each with a resolution of $720 \times 1280$ and a frame rate of 240. We follow (Chen et al., 2022; 2023b; Xu et al., 2021) to split this dataset into 100 training, 16 validation, and 17 test sets. Upon frame extraction, we employed the widely-used event simulation method vid2e (Gehrig et al., 2020), which accounts for real noise distribution, ensuring robust neural network training with enhanced generalization. In generating the inputs and ground truth of the network, we adopted and extended the previous works (Chen et al., 2023b; 2022) to accommodate multi-frame input. Specifically, input and ground truth (GT) frames are selected via sliding windows. We define the window size as $W$, the number of input frames as $N_{in}$, and the interval as $S$. They interrelate as: $W = (N_{in} - 1) * (S + 1) + 1$ For instance, with 4 frames input at 7-frame intervals, 25 frames are chosen. The 1st, 9th, 17th, and 25th frames become the input after down-sampling, while frames 1-25 serve as the GT. We adopted two strategies in line with prior works (Chen et al., 2022; 2023b): **I)** A fixed magnification set at $4\times$ the input resolution, and **II)** A variable enlargement strategy, wherein the scaling factor is governed by a $\mathcal{U}(1, 8)$ distribution.

**2) GoPro Dataset (Nah et al., 2017):** Both the GoPro and Adobe240 datasets share a resolution of $720 \times 1280$ and a frame rate of 240 $fps$. However, the GoPro dataset is more compact, encompassing 22 training videos and 11 for testing. Owing to this, prior research (Chen et al., 2023b; 2022) predominantly utilized the GoPro test set for quantitative evaluations, neglecting its training set. In the interest of fairness, we adopted the same approach.

**3) BS-ERGB (Tulyakov et al., 2022):** This dataset, captured with a spectroscope, comprises paired events and RGB frames from real-world scenarios. It has a resolution of $970 \times 625$ with RGB frames captured at 28 fps. Of the 123 videos in the dataset, 47 are allocated for training, 19 for validation, and 26 for testing. Each video contains between 100 to 600 frames. Despite being a real-world dataset, its size is inadequate for standalone training of a frame interpolation network. In previous work, *e.g.*TimeLens++ (Tulyakov et al., 2022), the Vime90k (Xue et al., 2019) simulation dataset was initially utilized for pre-training, followed by fine-tuning using this real-world dataset. We, on the other hand, opted for a model pre-trained on Adobe240 for our fine-tuning.

**4) CED dataset (Scheerlinck et al., 2019a):** This is another real-world dataset in SR research where both frames and events exhibit a resolution of $346 \times 260$. Following the previous research (Jing et al., 2021), we conducted preprocessing on this dataset.

### C.2  More Details about Implementation Details

Our experiments encompass four datasets, compared against a variety of benchmark methods. To guarantee fairness in these comparisons, we provide additional details regarding our experimental approach. For the Adobe240 and GoPro datasets, our training and testing protocols adhere to methods established in prior research (Chen et al., 2023b; 2022). Specifically, our approach diverges from prior studies in handling the smaller-scale BS-ERGB dataset. Unlike previous methods that pre-trained on the larger Vime90k dataset before fine-tuning on BS-ERGB, we opted for a different strategy. Our method involves pre-training on the Adobe240 dataset. During this phase, we input

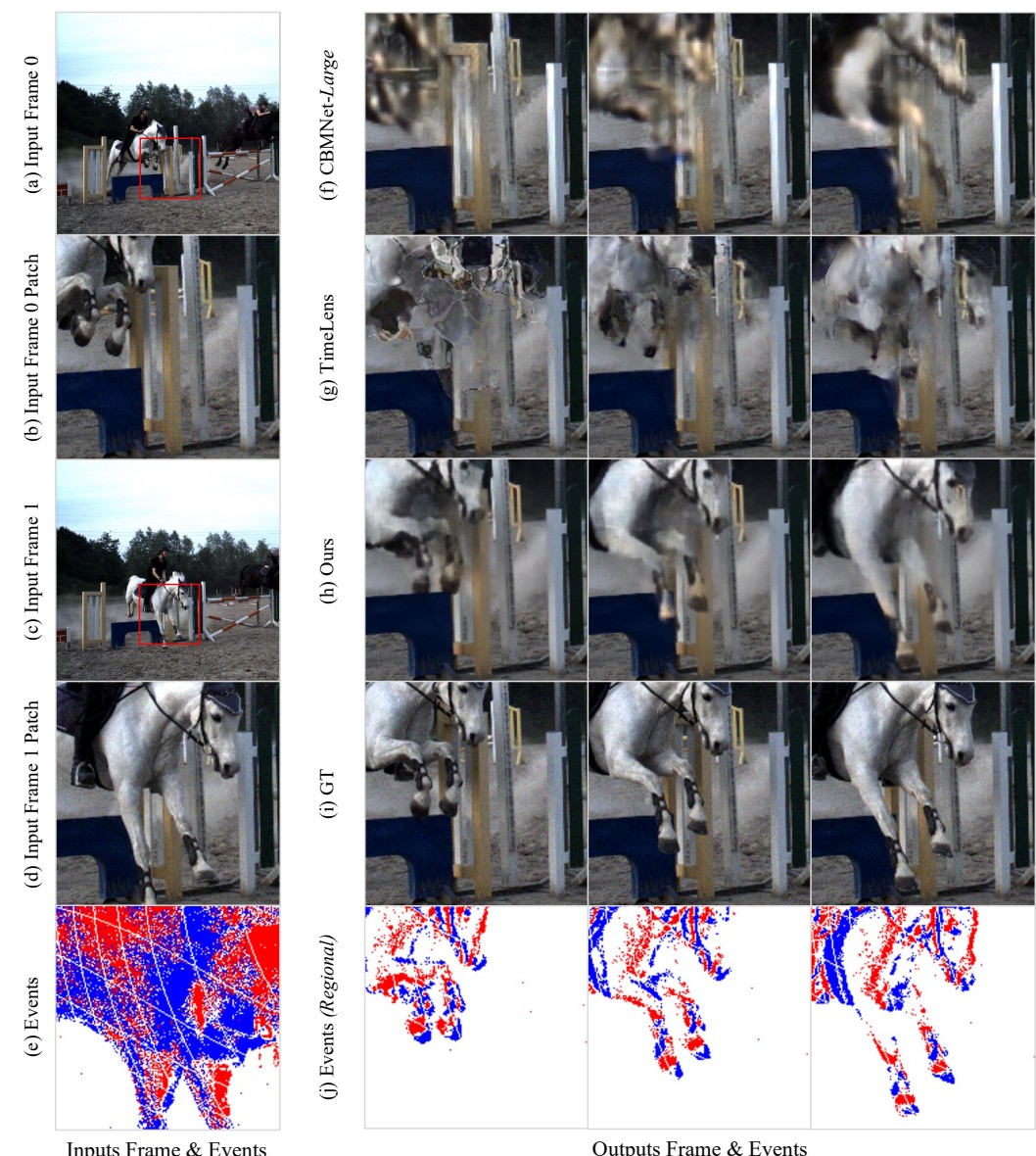

Figure 7: More visualization results on real-world data set (Tulyakov et al., 2021).

four video frames, with a gap of seven frames between each pair. From a total of 25 frames, 20 are randomly chosen as the ground truth. In the pre-training phase, our focus is solely on modules related to frame insertion, omitting any upsampling procedures. Consequently, the spatial embedding aspect is excluded, and the decoding is performed using only a four-layer MLP decoder. For the super-resolution experiments on the CED dataset, we deviated from pre-training and instead trained our model exclusively on the CED dataset. A more detailed comparison of the $2\times$ VSR is shown in Tab. 10. This approach was chosen to maintain a fair comparison with other works (Lu et al., 2023; Jing et al., 2021).

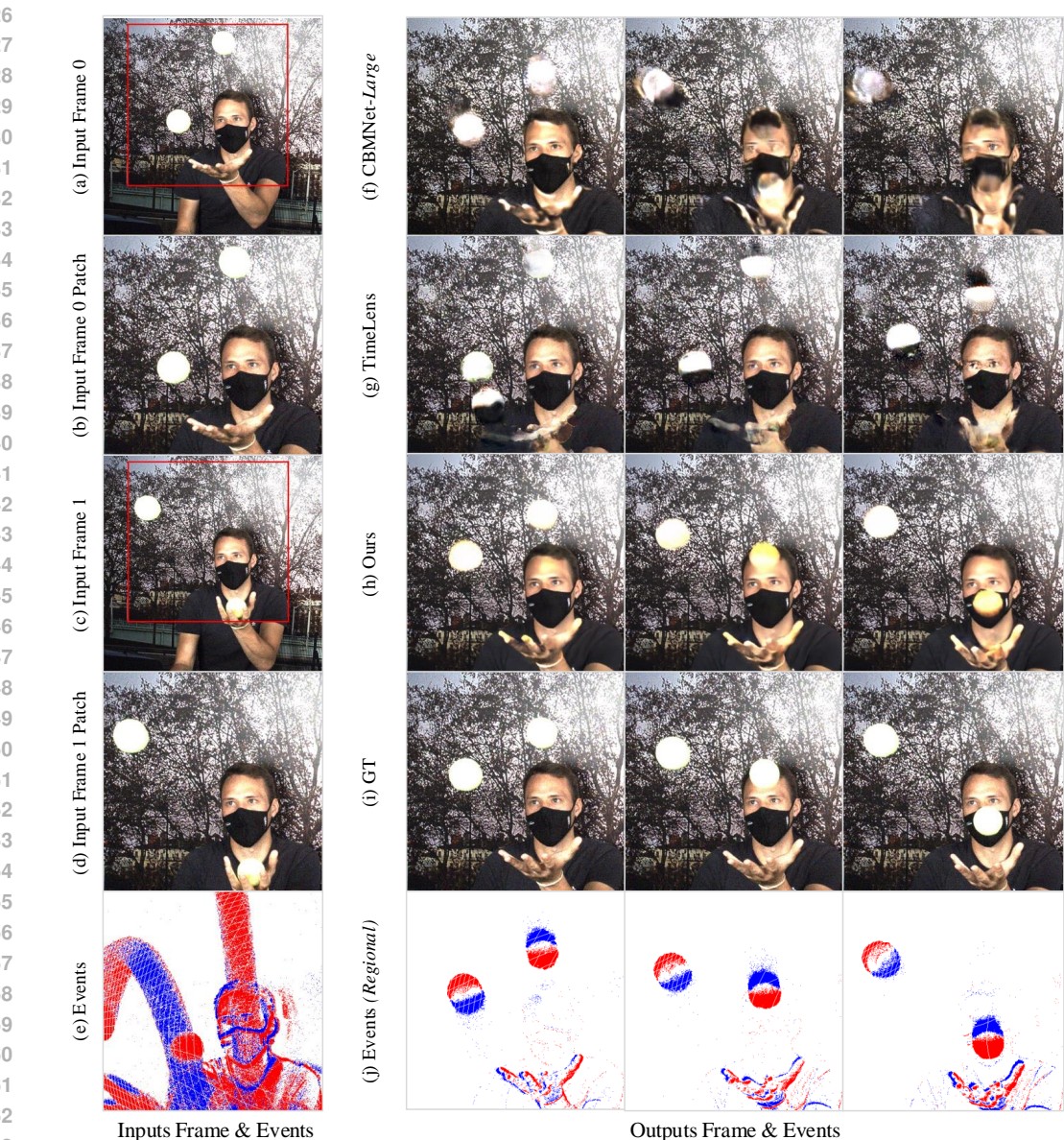

Figure 8: More visualization results on real-world data set (Tulyakov et al., 2021)

# D MORE EXPERIMENTS RESULTS AND ANALYSIS

## D.1 MORE VISUALIZATION WITH COMPARE METHODS

The visual comparisons provided in Fig.7, 8, and 9 illustrate the superiority of our method in recovering large-scale and long-distance motions across a variety of real-world scenarios. These results demonstrate the efficacy of our approach in fusing regional and holistic information to address complex motion dynamics, while also reducing ghosting artifacts that are commonly present in competing methods.

**Fig.7 (Localized Long-Distance Motion):** Our method excels in reconstructing fine-grained motion details, such as the movement of the horse's legs. Unlike CBMNet and TimeLens, which either fail to capture the intricate details or introduce significant motion blur, our approach accurately restores the distinct positions of the horse's hooves. This capability stems from the integration of local

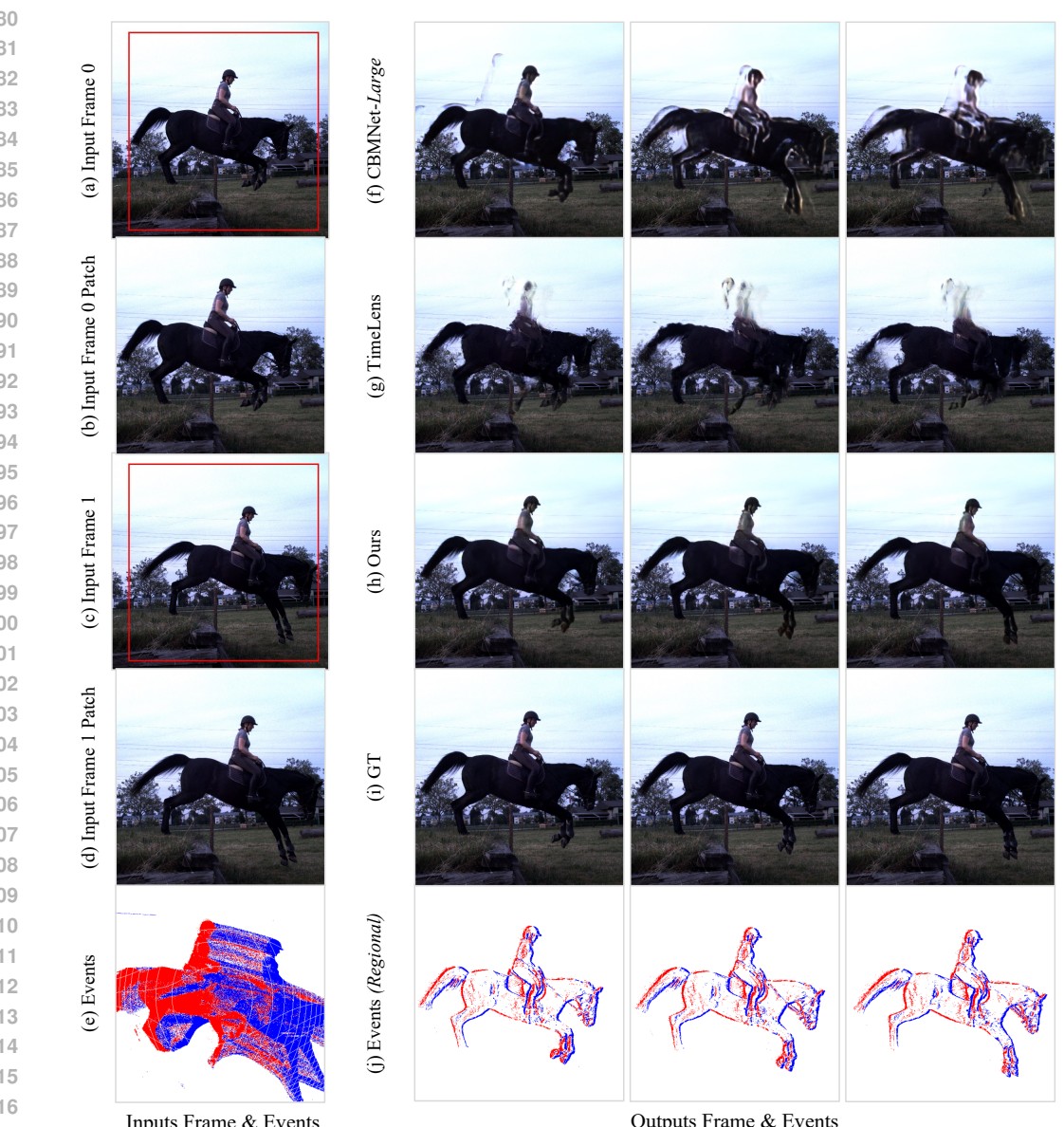

Figure 9: More visualization results on real-world data set (Tulyakov et al., 2021)

motion features captured by the regional branch and long-term temporal dependencies modeled by the holistic branch.

**Fig.8 (Small Object Long-Distance Motion):** In this example, where the subject is juggling, the motion of small, fast-moving objects (balls) is challenging to capture. Our method successfully tracks the trajectory of each ball, producing sharp and well-aligned results. Competing methods exhibit noticeable ghosting and fail to preserve the spatial consistency of the balls. This demonstrates our model's ability to handle small-scale, high-speed motion effectively by leveraging its dual-branch feature extraction.

**Fig.9 (Large Object Long-Distance Motion):** For large-scale dynamic motion, such as the horse and rider jumping, our method demonstrates clear advantages. The horse's silhouette and rider's position are reconstructed with remarkable clarity and consistency. In contrast, competing methods introduce significant blurring and fail to preserve the integrity of the subject's shape, highlighting their limitations in managing large-scale motion over time.

Across all scenarios, our method generates outputs with **fewer ghosting artifacts**, as evidenced in the reduced double-exposure effects visible in the reconstructed frames. The integration of regional and holistic features enables our approach to capture both short-term and long-term motion dependencies, resulting in more realistic and temporally coherent video outputs. These results substantiate the robustness and generalizability of our method across diverse motion patterns and scales.

## D.2 Capturing Millisecond Motion with Event TPR

In the main paper, we conducted Ablation Studies to validate the effectiveness of the Temporal Pyramid Representation (TPR) in capturing both short- and long-term motion dynamics. To further analyze and understand the feature extraction process, this section provides a visualization of TPR's capability to capture local motion dynamics in extreme temporal conditions. Specifically, we explore the regional and holistic features derived from the event data, showcasing their complementary strengths in capturing motion and static information.

As shown in Figure 10, the event data provides a unique advantage in capturing rapid, millisecond-scale motion that cannot be effectively represented by frame-based methods alone. For instance, the regional features derived from TPR (Fig. 10 **(g)**) focus on localized, short-term motion, while the holistic features (Fig. 10 **(f)**) encapsulate global, long-term context. This dual representation allows our method to effectively disentangle motion information from static background details.

Key Observations:

**Scenario 1: Traffic Scene with Localized Dynamics** In a scene where vehicles are moving in a stationary urban background, the regional features capture the intricate, millisecond-level motion of the vehicles, whereas the holistic features retain the overall scene structure, including static elements like traffic signs and buildings. This dual capability allows our method to handle dynamic scenes with both fast-moving and stationary elements seamlessly.

**Scenario 2: Basketball Player in Motion** In the visualization of a basketball player dribbling the ball, the Holistic Features predominantly encode static background details, such as trees and other stationary objects in the scene. Meanwhile, the Regional Features emphasize the dynamic movement of the basketball, effectively isolating the short-term motion signals caused by its rapid movement. This highlights TPR's strength in focusing on localized motion, even for objects moving at high speeds.

These visualizations provide strong evidence of TPR's capability to model fine-grained, short-term motion by leveraging its hierarchical structure. The event data complements frame-based information by focusing on temporal granularity, enabling the extraction of rich local motion features.

We encourage readers to refer to the supplementary videos, where these cases are further demonstrated, showcasing how TPR effectively integrates event and frame data to handle complex motion patterns. This visualization underscores the unique advantages of TPR as a novel event representation that combines regional and holistic information, enabling robust spatiotemporal super-resolution.

## D.3 Analysis of Input Frames and Long-distance Modeling

In the main paper, our ablation studies demonstrated that increasing the number of input frames significantly improves interpolation performance. Figure 11 compares the outputs of two models: one using two input frames (0, 1) and another using four input frames (-1, 0, 1, 2).

With two input frames, the model struggles to capture complex motion, as seen in the less accurate predictions of the basketball's trajectory in Fig.11 (b). In contrast, the four-frame model leverages additional temporal information to model longer-term motion, producing smoother and more accurate results, as shown in Fig.11 (c). The holistic feature visualizations further highlight the richer temporal dependencies captured by the four-frame model, allowing it to handle challenging motion scenarios more effectively.

This analysis underscores the strength of our method in utilizing **multi-frame inputs** to improve temporal coherence and enhance interpolation quality, particularly for long-distance or rapid motion.

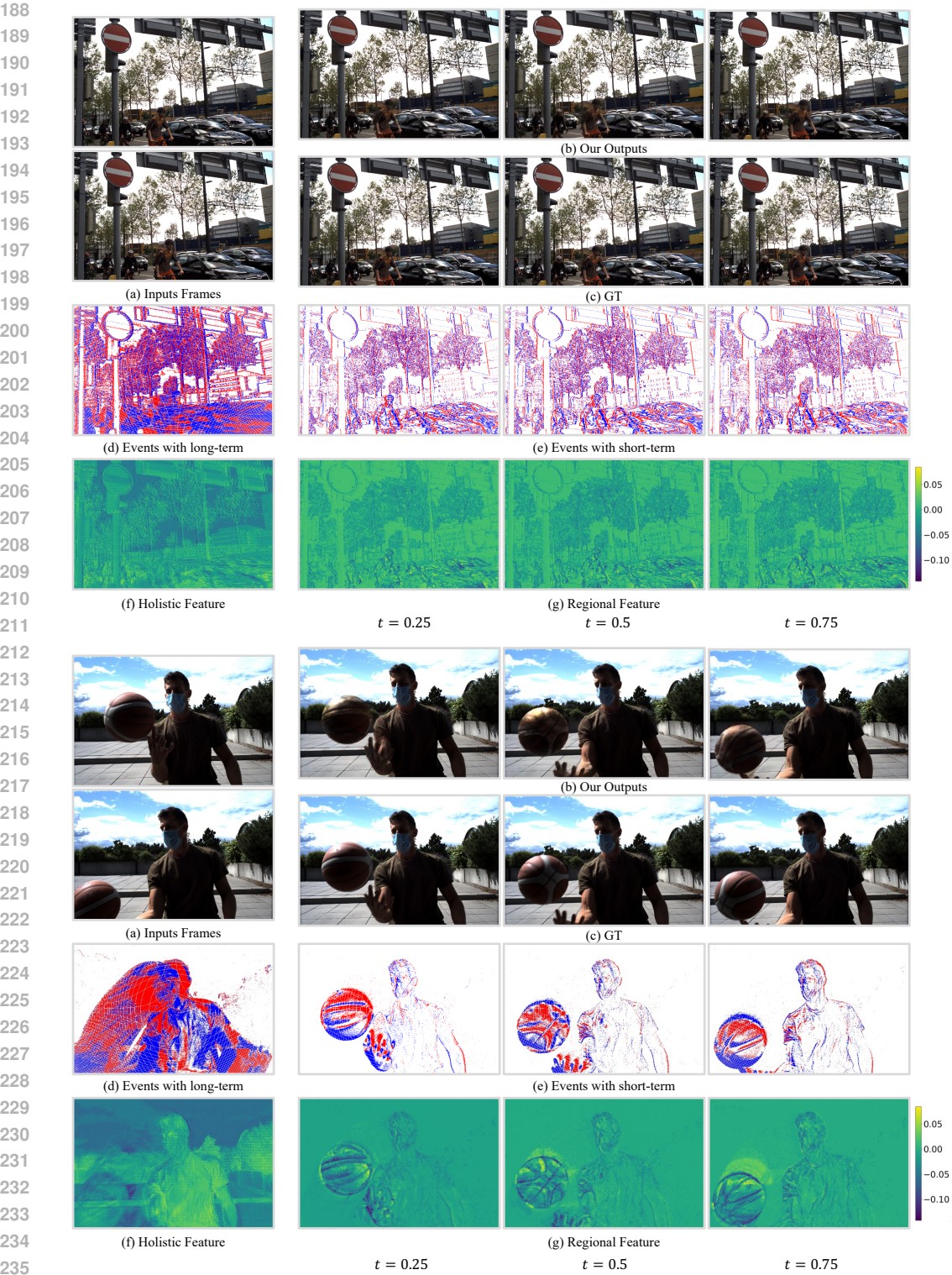

Figure 10: Feature visualization on real data (Tulyakov et al., 2022): **(f)** shows the holistic feature, $F^g$, derived from multiple frames and events; **(g)** depict the regional features ($F_t^g$), highlighting the capability to capture local motion.

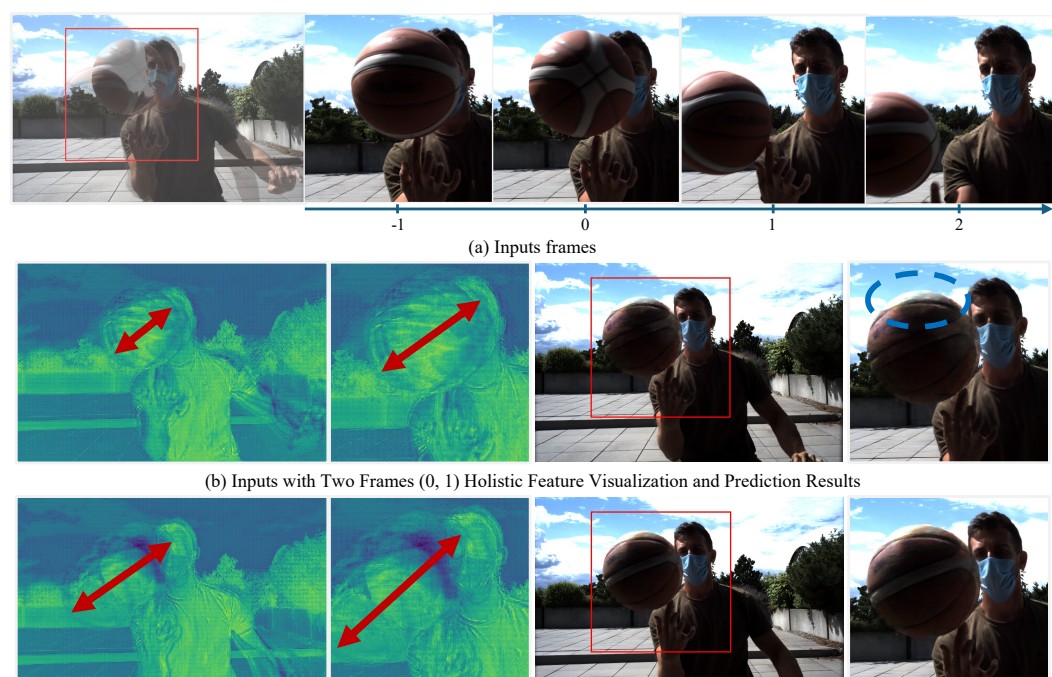

(a) Inputs frames

(b) Inputs with Two Frames (0, 1) Holistic Feature Visualization and Prediction Results

(c) Inputs with Four Frames (-1,0, 1,2) Holistic Feature Visualization and Prediction Results

Figure 11: Visualization of Holistic Features for multi-frame input.

## D.4 MORE METRIC EVALUATION RESULTS ON REAL-WORLD DATASET

Table 8: No-reference metric evaluation results on real-world dataset (Tulyakov et al., 2021) using Skip-3 testing. Lower values indicate better performance.

| Method | Average NIQE | Average PI |
|---|---|---|
| TimeLens | 4.6855 | 3.9610 |
| CBMNet | 3.2006 | 2.8634 |
| Ours | **3.1483** | **2.6397** |

Table 8 compares the perceptual quality of TimeLens, CBMNet, and our method on the real-world dataset using NIQE and PI metrics. Both metrics evaluate image quality without reference to ground truth, making them ideal for real-world scenarios.

**NIQE:** Our method achieves the lowest NIQE score of **3.1483**, indicating the most natural-looking outputs. While CBMNet performs significantly better than TimeLens, its score of **3.2006** is slightly higher than ours, showing the advantages of our approach in restoring naturalness.

**PI:** Similarly, our method outperforms CBMNet and TimeLens in perceptual realism, with the lowest PI score of **2.6397**. The gap between our method and CBMNet (0.2237) highlights our improved capacity to generate realistic outputs, particularly in challenging real-world conditions.

These results demonstrate that our method effectively leverages SOTA event-based methods to surpass the competing approaches in generating both natural and perceptually realistic results.

## D.5 MORE EXPERIMENTS IN APLIX-VSR DATASET

The ALPIX-VSR (Lu et al., 2023) dataset consists of 26 videos, each containing aligned video frames and events. However, note that the event data in this dataset is not in a stream format but is in the form of event-images. Therefore, the Temporal Pyramid Representation (TPR) structure cannot be directly applied. Instead, for the regional branch, we use the closest event-image as input

Table 9: Quantitative comparison (PSNR/SSIM) of our methods and other methods on the ALPIX-VSR dataset. * denotes the values obtained from the official pre-trained models.

| Scale | Methods | PSNR | SSIM |
|---|---|---|---|
| ×2 | E-VSR | 36.10 | 0.9761 |
| | EG-VSR | 38.25 | 0.9822 |
| | Ours | **38.32** | **0.9891** |
| ×4 | E-VSR | 32.54 | 0.9163 |
| | BasicVSR++ | 35.30 | 0.9353 |
| | EG-VSR | 37.12 | 0.9503 |
| | Ours | **37.96** | **0.9682** |
| ×6 | VideoINR* | 31.15 | 0.9084 |
| | EG-VSR | 31.85 | 0.9267 |
| | Ours | **33.60** | **0.9421** |
| ×8 | VideoINR* | 28.11 | 0.8625 |
| | EG-VSR | 28.53 | 0.8901 |
| | Ours | **29.31** | **0.8922** |

for the regional branch. Following previous methods such as EG-VSR (Lu et al., 2023), we trained our model for 100 epochs.

As shown in Table 9, our model outperforms previous methods across all scales, and its advantage becomes more pronounced as the upscaling factor increases. At lower upscaling factors (e.g., 2×), the performance gap between our model and competing methods is modest but significant, reflecting our ability to recover fine-grained details. Specifically, we achieve an improvement of 0.07 dB in PSNR compared to EG-VSR, while also demonstrating better structural similarity, as shown by the higher SSIM values.

However, as the upscaling factor increases, the superiority of our method becomes more evident. For instance, at 4× upscaling, our approach surpasses EG-VSR by 0.84 dB in PSNR, and the SSIM improves significantly from 0.9503 to 0.9682. This indicates that our framework effectively addresses the challenges posed by higher resolutions, where maintaining temporal consistency and spatial detail becomes increasingly difficult for traditional methods.

At the extreme upscaling factors of 6× and 8×, the robustness of our method becomes particularly apparent. For 6× super-resolution, our method achieves a PSNR improvement of 1.75 dB over EG-VSR, and the SSIM increases from 0.9267 to 0.9421. Similarly, at 8× upscaling, our model outperforms EG-VSR by 0.78 dB in PSNR and achieves a higher SSIM, overcoming the limitations of competing approaches like VideoINR and demonstrating a clear advantage in preserving both visual quality and structural integrity under challenging conditions.

These results illustrate that our method not only excels at lower upscaling factors but also maintains its effectiveness as the resolution demands increase, thereby setting a new benchmark for event-guided video super-resolution. The improvements at higher scales underscore the strength of our architectural innovations, particularly in leveraging event-image inputs to effectively address long-term dependencies and fine-grained motion dynamics.

### D.6 STABILITY OF VFI IN VARIOUS TIMESTAMPS:

Our method demonstrates temporal stability, accurately estimating motion states at each time point during frame interpolation. Specifically, whether in the GoPro-*Average* and Adobe-*Average* of Tab. 1, or the 12-*skip* and 16-*skip* tests of Tab. 2, our method significantly outperforms previous methods by at least $1.2dB$ with $4\times$ VSR.

Fig. 12 shows the relationship between timestamps and PSNR, validating the greater stability of our method compared to VideoINR (Chen et al., 2022), especially around the $0.5s$ mark, where VideoINR experiences a notable decrease. This observation is also reflected in the visualization results of Fig. 1 and Fig. 5 in main paper and Fig. 13 and Fig. 14 in appendix. In Fig. 13 and Fig. 14, whether using real-world or simulated data, our method demonstrates superior temporal consistency, exhibiting fewer artifacts compared to previous methods such as VideoINR (Chen et al., 2022) and Time-

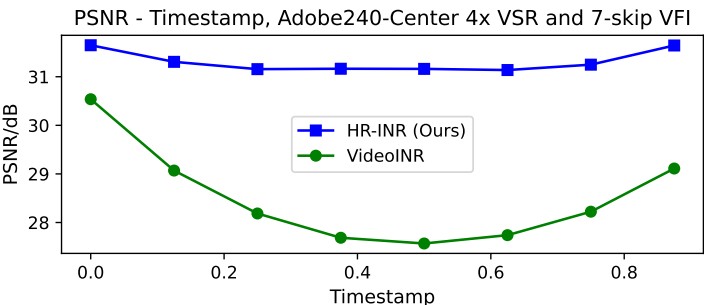

Figure 12: Timestamps and corresponding PSNR for each frame during $4\times$ VSR and 7-*skip* VFI on the Adobe240 dataset (Su et al., 2017).

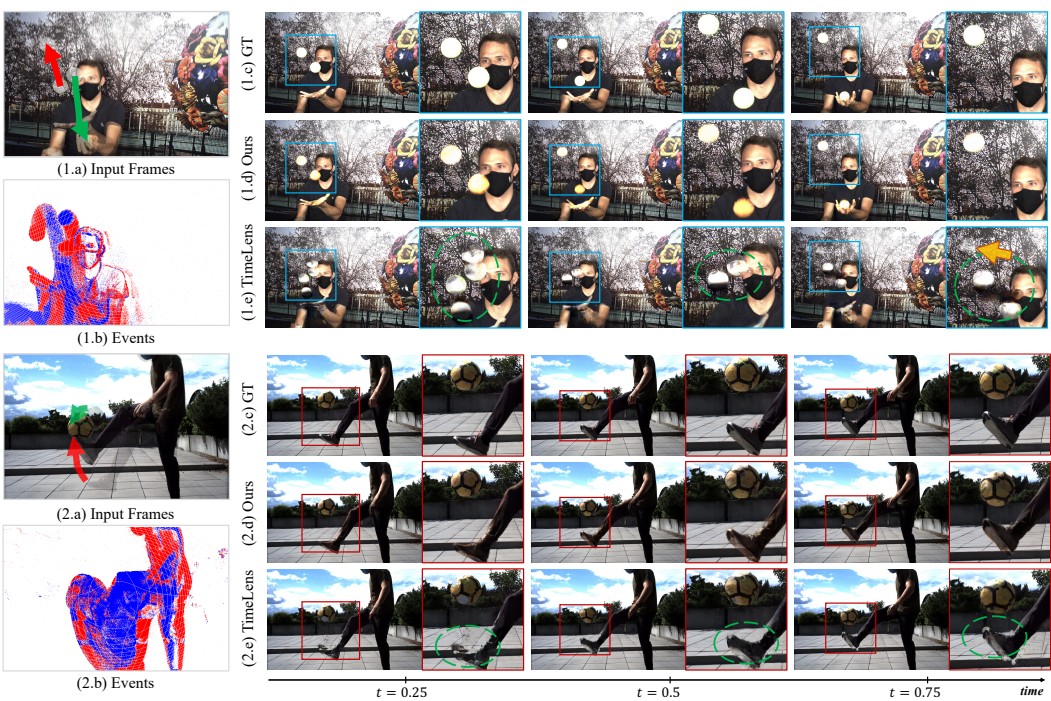

Figure 13: 3-*skip* frame interpolation visualization results in BS-ERGB dataset (Tulyakov et al., 2022). Our method accurately captures **local non-linear motion** (*e.g.*, balls in (1.d) and (2.d)), surpassing Time-Lens (Tulyakov et al., 2022), which exhibits ghosting and holes (green circles). A yellow arrow shows Time-Lens's inaccurate ball positioning.

Lens (Tulyakov et al., 2021). This showcases the real-world effectiveness of our method's temporal stability.

The main reason is that optical flow-based methods (Chen et al., 2022; 2023b; Tulyakov et al., 2021) suffer from instability in flow estimation at time **far from** the reference frames (at 0 and 1 timestamp), which impacts the motion estimation. In contrast, this limitation does not affect our method and maintains higher stability throughout the temporal sequence.

### D.7 VIDEO SUPER-RESOLUTION WITH $2\times$ IN CED DATASET

In Tab. 10, our proposed method demonstrates the highest performance across various clips in the CED dataset, significantly outperforming previous methods, including EG-VSR. Our approach consistently achieves higher PSNR and SSIM values, indicating superior visual quality and reconstruction accuracy. For instance, on average, our method improves PSNR by 3.32 $dB$ and SSIM by

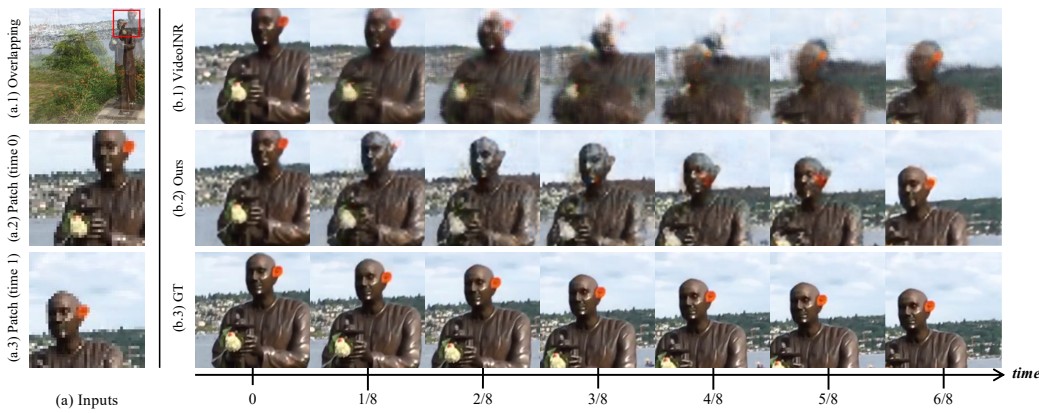

(a.1) Overlapping  (a.2) Patch (time 0)  (a.3) Patch (time 1)

(b.1) VideoINR  (b.2) Ours  (b.3) GT

(a) Inputs   0   1/8   2/8   3/8   4/8   5/8   6/8   time

Figure 14: $4\times$ VSR and 7-*skip* VFI visualization results. *Please refer to the supplementary material for the **video** of this case.*

Table 10: Quantitative results (PSNR/SSIM) of the proposed our framework and other methods on the CED dataset for $\times 2$. Because the official training code is not available, * denoted values were acquired from the pre-trained model that the authors have released.

| Clip Name | TDAN (Tian et al., 2020) | SOF (Wang et al., 2020b) | RBPN (Haris et al., 2019) | VideoINR* (Chen et al., 2022) | E-VSR (Jing et al., 2021) | EG-VSR (Lu et al., 2023) | Ours |
|---|---|---|---|---|---|---|---|
| people_dynamic_wave | 35.83 / 0.9540 | 33.32 / 0.9360 | 40.07 / 0.9868 | 27.47 / 0.8229 | 41.08 / 0.9891 | 38.78 / 0.9794 | 41.50 / 0.9901 |
| indoors_foosball_2 | 32.12 / 0.9339 | 30.86 / 0.9253 | 34.15 / 0.9739 | 26.03 / 0.7766 | 34.77 / 0.9775 | 38.68 / 0.9750 | 42.17 / 0.9904 |
| simple_wires_2 | 31.57 / 0.9466 | 30.12 / 0.9326 | 33.83 / 0.9739 | 26.77 / 0.8321 | 34.44 / 0.9773 | 38.67 / 0.9815 | 42.21 / 0.9922 |
| people_dynamic_dancing | 35.73 / 0.9566 | 32.93 / 0.9388 | 39.56 / 0.9869 | 27.36 / 0.8202 | 40.49 / 0.9891 | 39.06 / 0.9798 | 42.02 / 0.9913 |
| people_dynamic_jumping | 35.42 / 0.9536 | 32.79 / 0.9347 | 39.44 / 0.9859 | 27.24 / 0.8183 | 40.32 / 0.9880 | 38.93 / 0.9792 | 42.09 / 0.9916 |
| simple_fruit_fast | 37.75 / 0.9440 | 37.22 / 0.9390 | 40.33 / 0.9782 | 27.21 / 0.8456 | 40.80 / 0.9801 | 41.96 / 0.9821 | 43.96 / 0.9912 |
| outdoor_jumping_infrared_2 | 28.91 / 0.9062 | 26.67 / 0.8746 | 30.36 / 0.9648 | 26.88 / 0.8226 | 30.70 / 0.9698 | 38.03 / 0.9755 | 42.68 / 0.9902 |
| simple_carpet_fast | 32.54 / 0.9006 | 31.83 / 0.8774 | 34.91 / 0.9502 | 24.21 / 0.5909 | 35.16 / 0.9536 | 36.14 / 0.9635 | 39.80 / 0.9853 |
| people_dynamic_armroll | 35.55 / 0.9541 | 32.79 / 0.9345 | 40.05 / 0.9878 | 27.26 / 0.8193 | 41.00 / 0.9898 | 38.84 / 0.9787 | 41.99 / 0.9915 |
| indoors_kitchen_2 | 30.67 / 0.9323 | 29.61 / 0.9192 | 31.51 / 0.9551 | 26.44 / 0.7502 | 31.79 / 0.9586 | 37.68 / 0.9726 | 41.61 / 0.9901 |
| people_dynamic_sitting | 35.09 / 0.9561 | 32.13 / 0.9367 | 39.03 / 0.9862 | 27.63 / 0.8230 | 39.97 / 0.9884 | 38.86 / 0.9810 | 41.99 / 0.9917 |
| average PSNR/SSIM | 33.74 / 0.9398 | 31.84 / 0.9226 | 36.66 / 0.9754 | 26.77 / 0.7938 | 37.32 / 0.9783 | 38.69 / 0.9771 | **42.01 / 0.9905** |

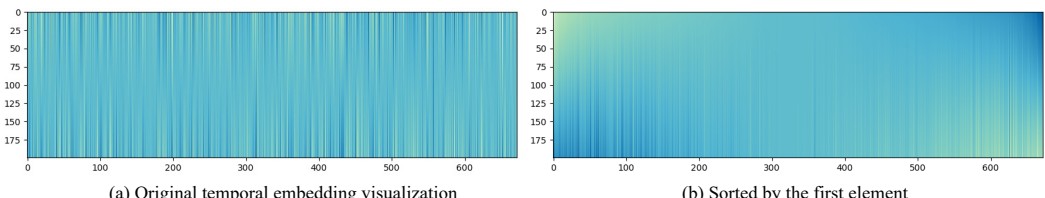

(a) Original temporal embedding visualization          (b) Sorted by the first element

Figure 15: Visualization of Temporal Embedding. Figure (a) shows the original visualization of Temporal Embedding, while figure (b) displays the results after sorting. The sorting is based on the size of the first element, arranged in ascending order. We use MLP decoding, where the order is not crucial. However, to more clearly demonstrate the outcomes of Temporal Embedding learning, we have chosen to present the sorted results.

0.0134 compared to EG-VSR. These results highlight the effectiveness of our framework in handling complex scenes, showcasing its robustness and reliability in video super-resolution tasks.

## D.8 TIME EMBEDDING FEATURE VISUALIZATION

Fig.15 illustrates the visualization of time embedding features. Compared to traditional sine-cosine embedding features, the learning-based approach performs better, as shown in Tab. 5. The visualized learning-based embedding features not only demonstrate the capability to learn periodic positional representations but also provide a richer expression of exposure information.

## Comparison of Total and Average Time for 34x Frame Interpolation

Figure 16: Comparison of total and average time for $34\times$ frame interpolation by different methods. Our method takes less time than TimeLens (Tulyakov et al., 2021), but slightly more time than VideoINR (Chen et al., 2022).

### D.9 BAD CASE ANALYSIS

We have observed that our method has certain limitations in some cases (Fig. 14). For example, when restoring color information, although our model can accurately reconstruct the contours of objects, the color information is often distorted or missing. This issue primarily arises due to the lack of color information in the event stream. We believe that with future advancements in color event technology, this problem will be effectively addressed.

### D.10 INFERENCE TIME ANALYSIS

In Fig. 15, we analyze the inference times of three different methods. Both our method and VideoINR achieve an average frame time of less than $100\ ms$ for $34\times$ frame interpolation. In contrast, TimeLens (Tulyakov et al., 2021) has an average frame time of $187\ ms$, which is more than double that of our method. The tests were conducted on a high-performance computer, and each method was tested 30 times, with the final inference time being the average of these 30 trials.

## E MORE VISUALIZATION RESULTS

Additional videos have been included in the supplementary materials to provide a more comprehensive demonstration of our method's visual results. Below, we enumerate these videos and briefly describe their key features. We then present more visualizations to demonstrate the generalization of our method on real data.

- `1-Adobe240:` This video contains the following five clips.
  - `IMG_0013-7skip4xsr-Cyclist:` In this video, the camera and the **people in the background** are in motion, creating a complex scene. Our method successfully recovers the **locally moving bicycle**, demonstrating exceptional video frame interpolation and super-resolution capabilities.
  - `IMG_0037-7skip4xsr-TrafficIntersectionManyCars:` The video demonstrates a **camera with slight movement**, capturing a **busy intersection bustling with vehicles**. Our method is capable of accurately recovering vehicles in motion within the scene, including the intricate details of rotating tires.
  - `IMG_0037a-7skip-MovingForegroundAndBackground:` The video includes both **distant and close-up elements**. In the close-up scenes, the comparative methods resulted in significant deformations and distortions.
  - `IMG_0045-7skip-PortraitSculpture:` This video demonstrates the effects under **significant**

**camera movement**. When the camera moves rapidly, frame-based methods tend to underperform.

- IMG_0175-7skip4xsr-LawnAndCar: The same scene occurs when the **camera moves violently**.
- IMG_0175-7skip4xsr-TreeComplexTexture: This video captures leaves, demonstrating that methods based on optical flow tend to fail in the presence of complex textures.

• 2-TimeLensPP-Ours-1: This video shows the performance of our method on real-world data sets and the visualization of features. Demonstrates that we effectively capture local motion.

• 3-Our-vs-Timelens: This video shows the results of comparing our method with Timelens (Tulyakov et al., 2021).

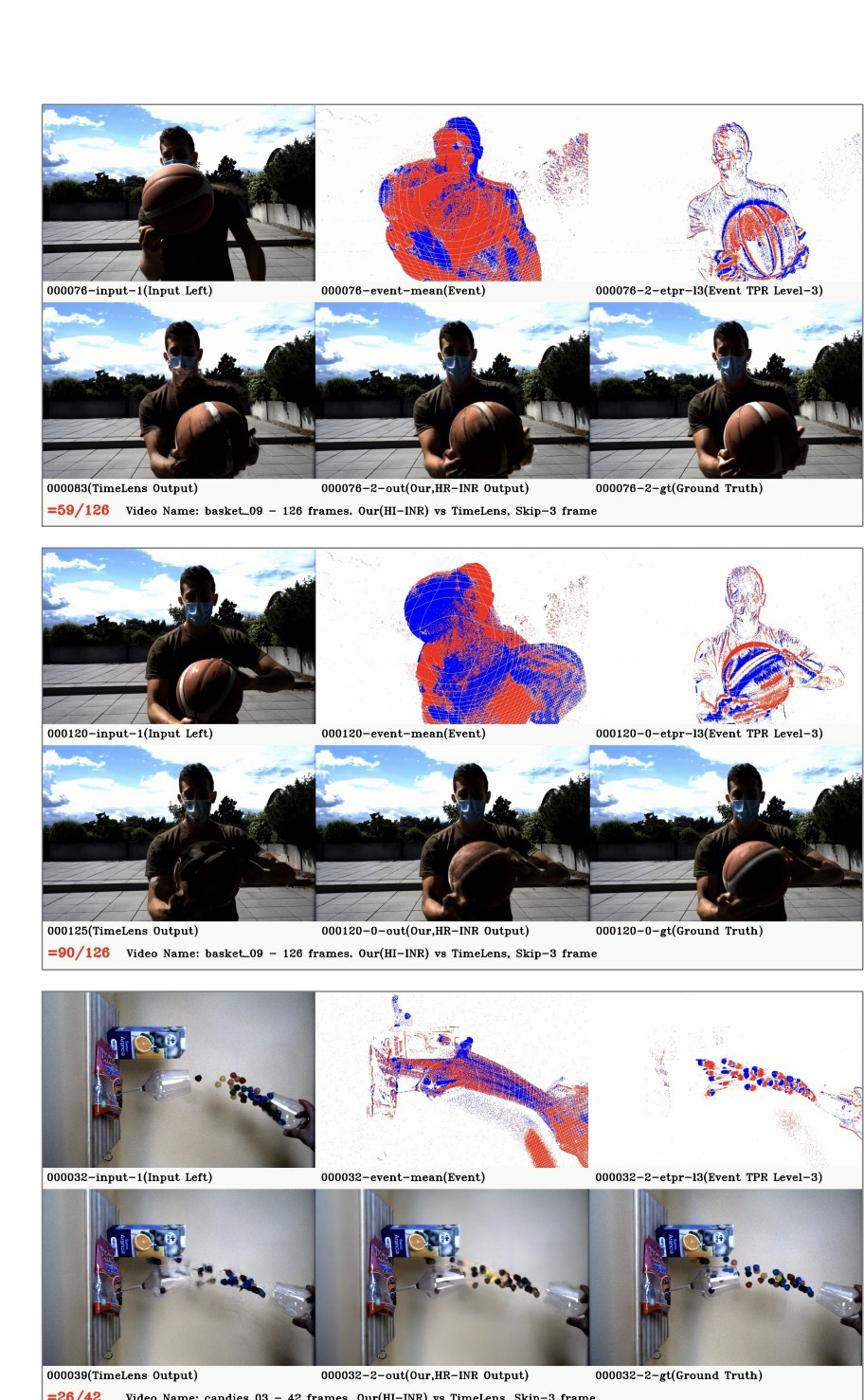

Figure 17: More visualization results on real-world data set (Tulyakov et al., 2021).

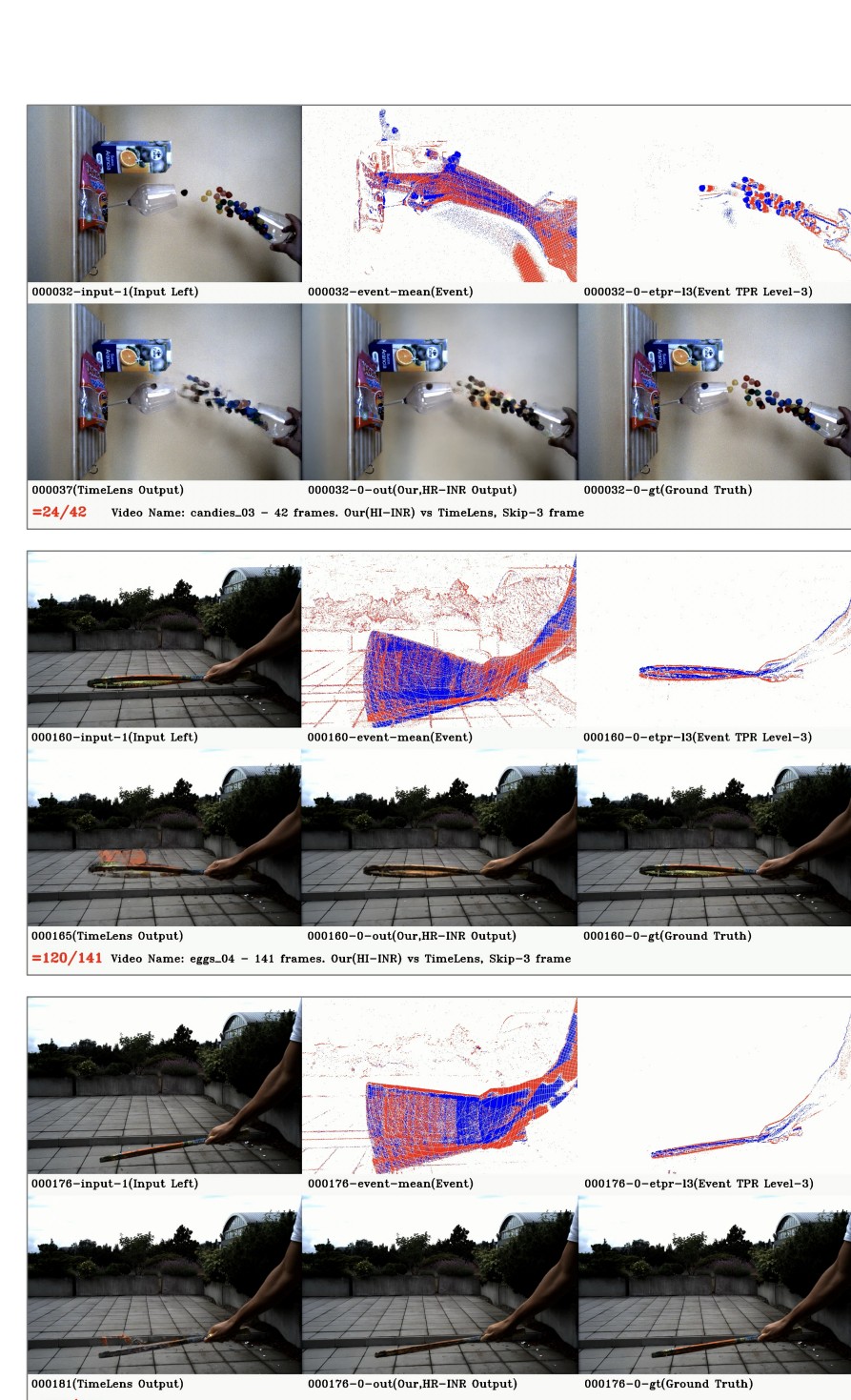

Figure 18: More visualization results on real-world data set (Tulyakov et al., 2021).

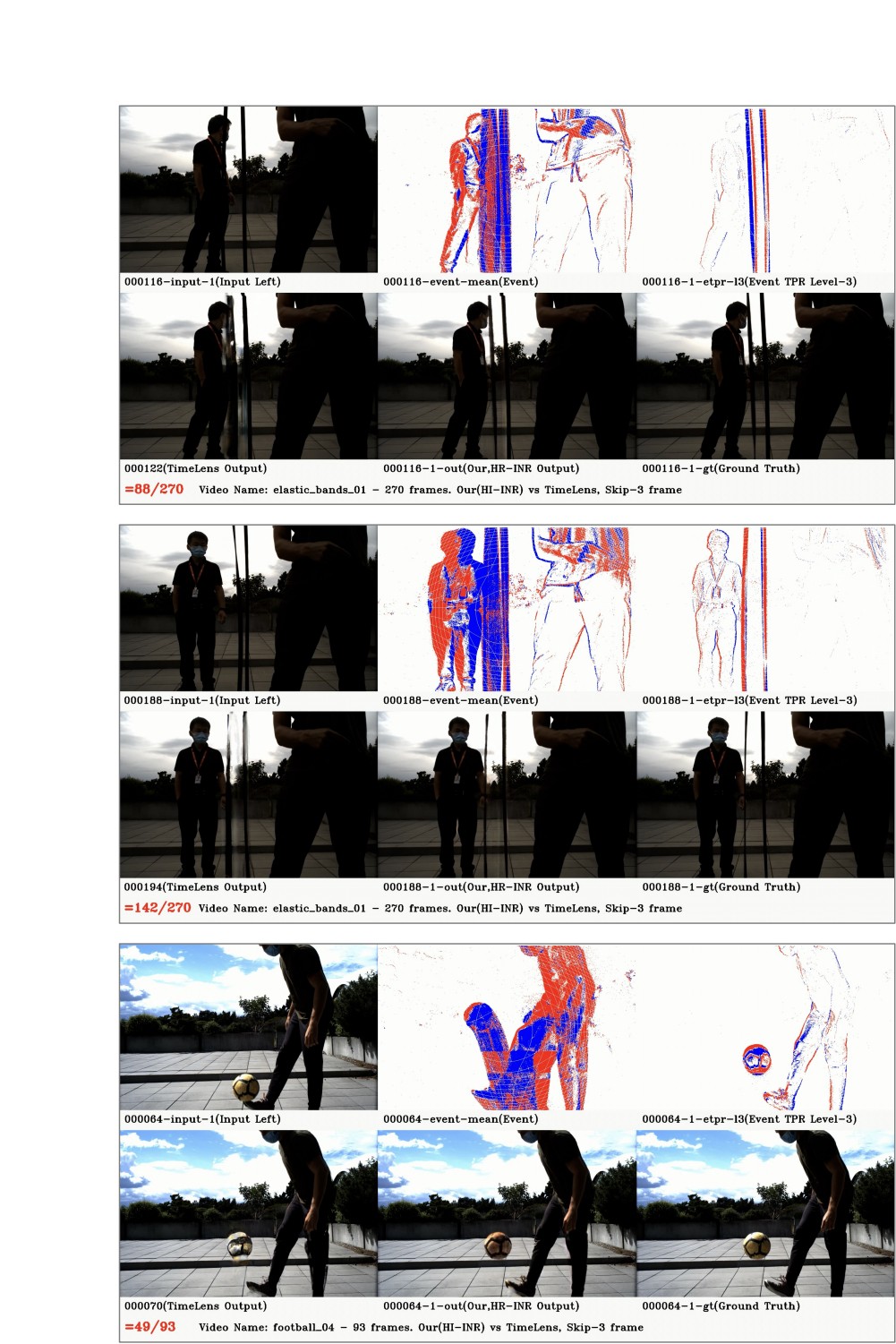

Figure 19: More visualization results on real-world data set (Tulyakov et al., 2021).

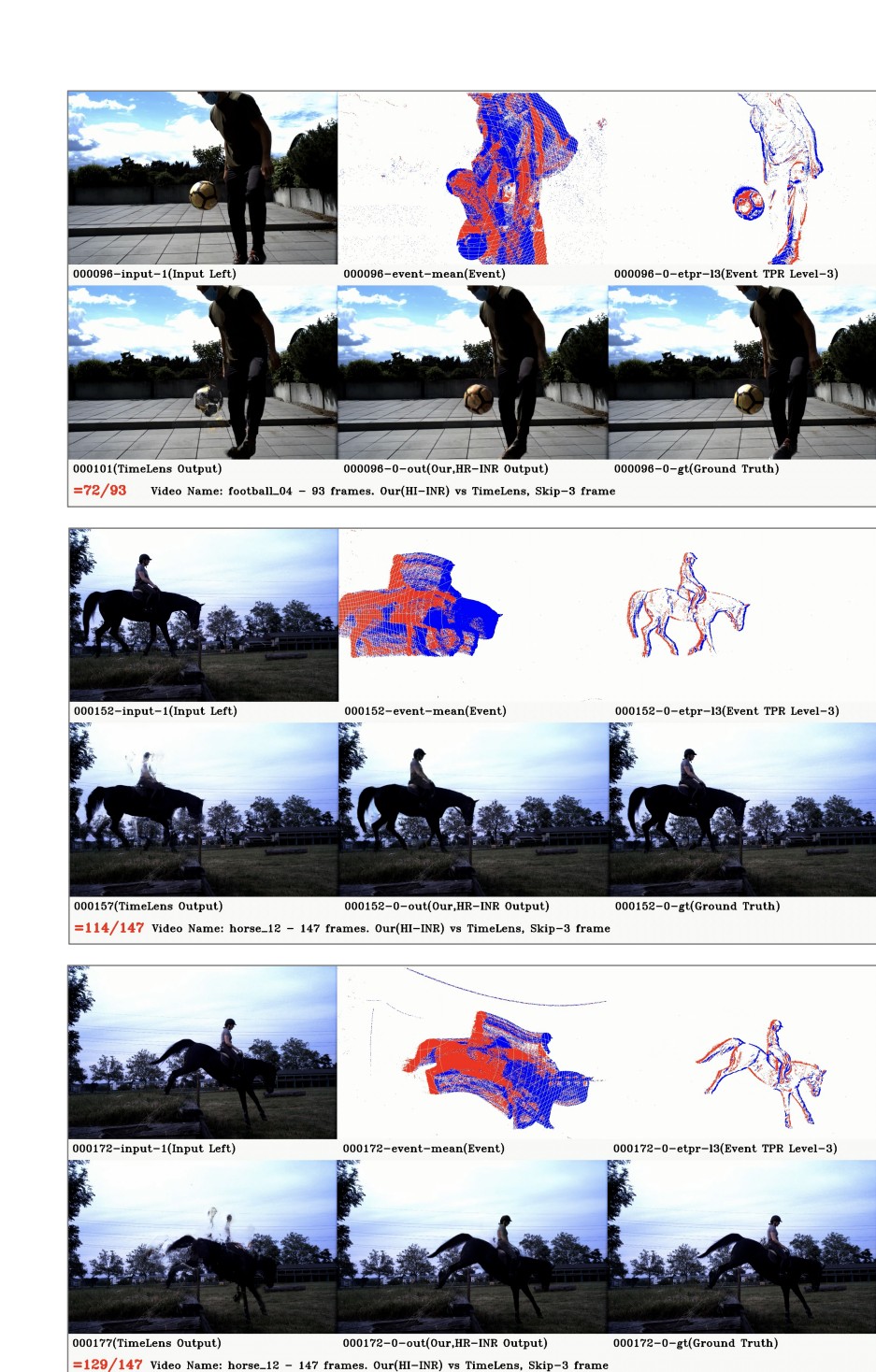

Figure 20: More visualization results on real-world data set (Tulyakov et al., 2021).

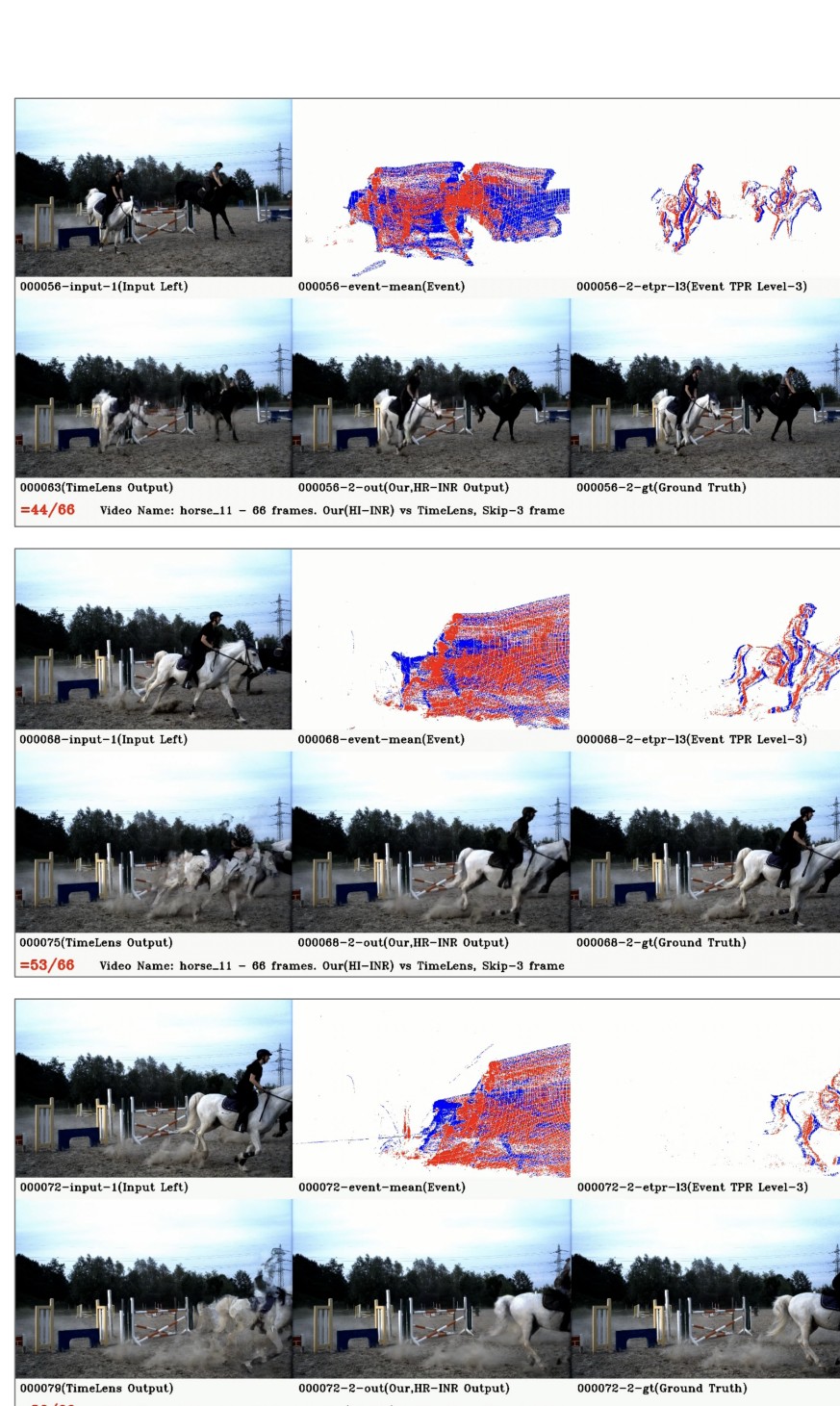

Figure 21: More visualization results on real-world data set (Tulyakov et al., 2021).

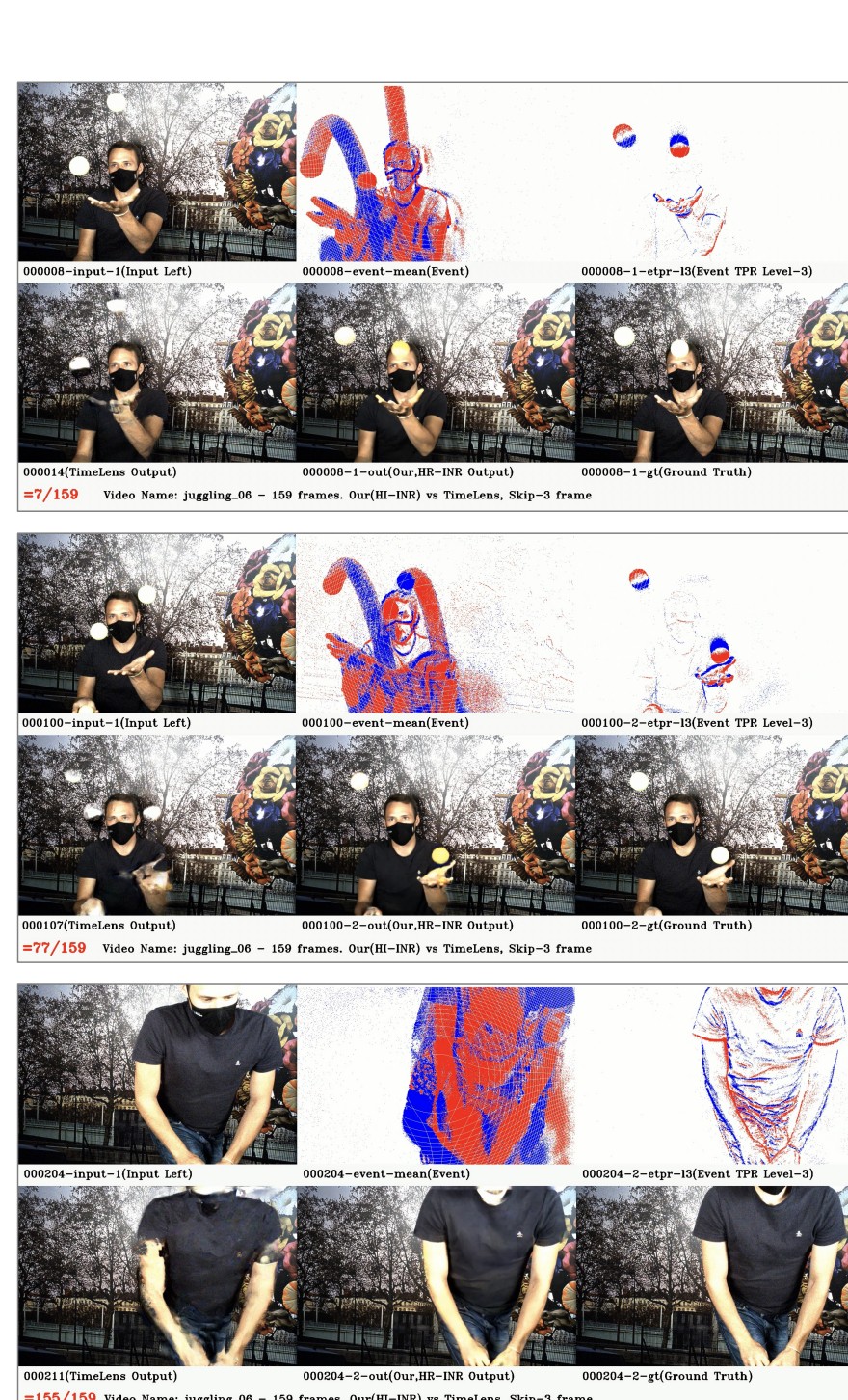

Figure 22: More visualization results on real-world data set (Tulyakov et al., 2021).

