# OpenReview forum: "Continuous Space-Time Video Super-Resolution via Event Camera"
_ICLR.cc/2025/Conference — Submitted to ICLR 2025_

### Official Review · Reviewer_fyeD · 2024-10-27

**Soundness:** 3
**Presentation:** 3
**Contribution:** 2
**Rating:** 5
**Confidence:** 4

**Summary:**

This paper proposes an event-guided method for simultaneously enhancing video resolution and frame rate at an arbitrary scale. It captures both holistic dependencies and regional motions based on INR. Experimental results show the superiority of the proposed method.

**Strengths:**

* The idea of simultaneously VSR and VFI using events is new. And the proposed method achieve the state-of-the-art (SOTA) performance on not only VSR but also VFI.
* The idea about using implicit neural representation (INR) could be new.

**Weaknesses:**

* Tech

  * The authors say they use INR for C-STVSR. However, as far as I know, INR is something like NeRF (\ie, using a neural network to represent something such as 3D/2D objects). However, in this paper the author do not define the INR in a clear manner. I cannot find where the INR is used. It seems that the term INR in this paper is about some feature layers used in the designed network architecture, not representations. It is much more like "designing a new network structure/proposing new modules" instead of propose a new INR. There are lots of works about event representation (for example,  PhasedLSTM [a],  MatrixLSTM [b] , NEST [c]). However, the authors do not mention, discuss, or compare with them.
  * How to define the regional and holistic events? It seems that they are just two handcrafted terms without clear meaning. The motivation of defining such terms is not clear.
  * It seems that the other event-based VFI methods can remove the motion blur. However, from Figure 1 we can see that the motion blur still exists.

* Experiments

  * The comparisons may be not that comprehensive enough. For example, the authors should compare with pipelines such as "SOTA event-based VFI method + SOTA event-based VSR method" (\eg,   CBMNet + EG-VSR and other combinations) in Table 1. Only in this way it can show that it achieves the SOTA performance on C-STVSR. It seems that the compared baselines do not involve such kinds of pipelines.
  * Why not evaluate on the GoPro and Adobe dataset for either VSR or VFI? It seems that the authors only evaluate the performance of C-STVSR on this two datasets. Besides, for VSR, why not evaluate the proposed method on ALPIX-VSR dataset in [d]?

* Writing

  * The writing quality is not that good. For example, the conference names in Ref are not unified (\eg, for CVPR, the authors use both "2019 IEEE/CVF Conference on Computer Vision and Pattern Recognition (CVPR) " and " Proceedings of the IEEE/CVF conference on computer vision and pattern recognition").

  [a] Phased LSTM: Accelerating recurrent network training for long or event-based sequences

  [b] M.: A differentiable recurrent surface for asynchronous event-based data

  [c] NEST: Neural Event Stack for Event-based Image Enhancement

  [d] Learning Spatial-Temporal Implicit Neural Representations for Event-Guided Video Super-Resolution

**Questions:**

* What is the upper bound of the number of FPS? This should be compared with the other VFI methods. Besides, the performance of setting different numbers of FPS should also be analyzed.
* In the abstract, the authors mention that continuous space-time video super-resolution (C-STVSR) aims to simultaneously enhance video resolution and frame rate at an arbitrary scale. Therefore, can the video be SR in any scale using the proposed method? Please show an example with a large scale (\eg, 8x 16x).

---

> ### Author Response · Authors · 2024-11-21
> **Response to Reviewer fyeD's Comments and Feedback (1/2)**
>
> # Author Response to Reviewer fyeD (1/2)
>
> Dear Reviewer fyeD,
>
> Thank you for your appreciation of the ideas presented in our paper; your encouragement is highly motivating. We have provided detailed responses to your questions below.
>
> ### Q.1 Clarification on the use and definition of INR
> > 1.1 The authors say they use INR for C-STVSR. However, as far as I know, INR is something like NeRF (\ie, using a neural network to represent something such as 3D/2D objects). However, in this paper the author do not define the INR in a clear manner. I cannot find where the INR is used. It seems that the term INR in this paper is about some feature layers used in the designed network architecture, not representations. It is much more like "designing a new network structure/proposing new modules" instead of propose a new INR. There are lots of works about event representation (for example, PhasedLSTM [a], MatrixLSTM [b] , NEST [c]). However, the authors do not mention, discuss, or compare with them.
>
> We appreciate your insightful feedback. We use the term Implicit Neural Representation (INR) following the conventions established by notable works such as VideoINR and MoTIF. Specifically, our model learns a function of time t and space s, as presented in Equation 1 of our paper. During inference, the time t and spatial coordinates s can be specified by the user. Thus, our model effectively learns an INR that maps continuous spatiotemporal coordinates to corresponding video content. We hope this clarifies our use of INR.
>
> Furthermore, thank you for bringing to our attention the relevant works on learning-based event representations, such as PhasedLSTM [a], MatrixLSTM [b], and NEST [c]. We will include discussions and comparisons with these studies in the revised version of our paper. Your valuable suggestions will significantly enhance the quality of our work.
>
> ### Q.2 Definitions and motivations for 'regional' and 'holistic' events
> > 1.2 How to define the regional and holistic events? It seems that they are just two handcrafted terms without clear meaning. The motivation of defining such terms is not clear.
>
> Thank you for highlighting this point. In our work, 'regional events' refer to event information localized in time, specifically captured by our Temporal Pyramid Representation (TPR). 'Holistic events' encompass global event information over a longer temporal duration. Our primary motivation is to leverage both local (short-term) and global (long-term) event information to effectively address challenges in interpolation and super-resolution involving rapid motions and long-term dependencies. We will clarify these definitions and their motivations more explicitly in the revised paper.
>
> ### Q.3 Motion blur remains in results despite using event-based VFI methods
> > 1.3 It seems that the other event-based VFI methods can remove the motion blur. However, from Figure 1 we can see that the motion blur still exists.
>
> We appreciate your observation. The interpolation methods we compared with, such as TimeLens, TimeLens++, and CBMNet, are designed for interpolation but do not simultaneously perform deblurring. While there are studies [1, 2, 3] that address both interpolation and deblurring, they fall under a different category of methods. Thanks for you insights. We will explore joint deblurring and interpolation in future research.
>
> ### Q.4 Comprehensive comparisons with combined SOTA methods
> > 2.1 The comparisons may be not that comprehensive enough. For example, the authors should compare with pipelines such as "SOTA event-based VFI method + SOTA event-based VSR method" (\eg, CBMNet + EG-VSR and other combinations) in Table 1. Only in this way it can show that it achieves the SOTA performance on C-STVSR. It seems that the compared baselines do not involve such kinds of pipelines.
>
> Thank you for your suggestion. We will include additional experiments comparing our method with combined pipelines of state-of-the-art event-based VFI and VSR methods, such as CBMNet + EG-VSR, in the revised paper. This will provide a more comprehensive evaluation of our method's performance on C-STVSR.
>
> ### Q.5 Evaluation on additional datasets
> > 2.2 Why not evaluate on the GoPro and Adobe dataset for either VSR or VFI? It seems that the authors only evaluate the performance of C-STVSR on these two datasets. Besides, for VSR, why not evaluate the proposed method on ALPIX-VSR dataset in [d]?
>
> Thank you for this valuable suggestion. We will include evaluations of our method on the GoPro and Adobe datasets for VSR and VFI tasks, as well as on the ALPIX-VSR dataset [d], in the revised paper. This will help demonstrate the generalizability and robustness of our approach across diverse datasets.

---

> ### Author Response · Authors · 2024-11-21
> **Response to Reviewer fyeD's Comments and Feedback (2/2)**
>
> ### Q.6 Improvement of writing quality and reference formatting
> > 3. The writing quality is not that good. For example, the conference names in Ref are not unified (\eg, for CVPR, the authors use both "2019 IEEE/CVF Conference on Computer Vision and Pattern Recognition (CVPR) " and " Proceedings of the IEEE/CVF conference on computer vision and pattern recognition").
>
> We appreciate your feedback regarding the writing quality. We have thoroughly revised the paper to address these issues, ensuring consistency in reference formatting and enhancing overall clarity and readability. Your input is instrumental in improving the quality of our manuscript.
>
> ### Q.7 Upper bound of FPS and analysis at different settings
> > 4. What is the upper bound of the number of FPS? This should be compared with the other VFI methods. Besides, the performance of setting different numbers of FPS should also be analyzed.
>
> Thank you for your question. Our model learns a continuous representation of the video, accepting time t as input, which theoretically allows for infinite temporal resolution (i.e., unlimited FPS). Therefore, we can upsample the video to any desired frame rate. In our experiments, we matched the ground truth (GT) frame rate due to the fixed frame rates of the GT data. We will include additional experiments and analyses in the revised paper to demonstrate the model's upsampling performance at various FPS settings and compare it with other VFI methods.
>
> ### Q.8 Capability of arbitrary spatial super-resolution
> > 5. In the abstract, the authors mention that continuous space-time video super-resolution (C-STVSR) aims to simultaneously enhance video resolution and frame rate at an arbitrary scale. Therefore, can the video be SR in any scale using the proposed method? Please show an example with a large scale (\eg, 8x 16x).
>
> Indeed, our model supports super-resolution at arbitrary spatial scales. We will include examples with larger scaling factors, such as 8× and 16×, in the revised paper to demonstrate this capability. This will illustrate the flexibility and effectiveness of our approach in handling high-resolution outputs.
>
> We hope that our responses address your concerns satisfactorily. Your constructive feedback has been invaluable in helping us improve our work, and we are grateful for your contributions.
>
> Sincerely,
>
> ## Reference:
> - [1] Sun, Lei, et al. "Event-based frame interpolation with ad-hoc deblurring." Proceedings of the IEEE/CVF Conference on Computer Vision and Pattern Recognition. 2023.
> - [2] Song, Chen, Chandrajit Bajaj, and Qixing Huang. "DeblurSR: Event-Based Motion Deblurring under the Spiking Representation." Proceedings of the AAAI Conference on Artificial Intelligence. Vol. 38. No. 5. 2024.
> - [3] Song, Chen, Qixing Huang, and Chandrajit Bajaj. "E-cir: Event-enhanced continuous intensity recovery." Proceedings of the IEEE/CVF Conference on Computer Vision and Pattern Recognition. 2022.

---

> > ### Comment · Reviewer_fyeD · 2024-11-25
> > **I decide to keep my score after reading the rebuttal.**
> >
> > Considering the fact that the authors do not provide any results and analyses in the rebuttal period, I decide to keep my score.

---

### Official Review · Reviewer_wQx2 · 2024-11-02

**Soundness:** 3
**Presentation:** 3
**Contribution:** 3
**Rating:** 5
**Confidence:** 5

**Summary:**

This paper presents a space-time video super-resolution method using an event camera. The proposed approach utilizes events and video inputs, employing an event temporal pyramid representation, a comprehensive event-frame feature extractor, and spatiotemporal decoding to produce high-frame-rate, high-resolution videos. Experimental results demonstrate that this method significantly outperforms existing state-of-the-art techniques, particularly in terms of temporal consistency, as shown in the supplementary videos.

**Strengths:**

1.	The proposed temporal pyramid representation captures motion information at multiple pyramid levels, allowing the subsequent feature extraction module to access a range of detail from sharp but sparse features to blurrier yet denser details.
2.	Two specialized feature extractors for events and frames are introduced, each designed to capture information at distinct temporal resolutions.
3.	The spatial-temporal decoding module, which integrates temporal and spatial embeddings, enhances the method’s temporal and spatial consistency, outperforming existing space-time video super-resolution techniques.

**Weaknesses:**

1.	The feature extractors appear to be based on a Swin-Transformer and U-Net (Swin-Unet) combination, which raises concerns about novelty; additional ablation studies would help validate their effectiveness.
2.	Given that this is a space-time video super-resolution approach, including one or two video metrics to assess quality and temporal consistency would provide a more comprehensive evaluation.
3.	The visual comparison results include only one or two baseline method, while more methods are compared in the tabular results.

**Questions:**

It appears that the proposed method can handle a longer sequence of events as input. In comparison, what is the length of the event input used by other methods in this study? Given that a long event sequence may not be suitable for other methods to provide optimal results, testing different sequence lengths would be beneficial.

---

> ### Author Response · Authors · 2024-11-21
> **Response to Reviewer wQx2's Comments and Feedback**
>
> Dear Reviewer wQx2,
>
> Thank you for your insightful comments and for recognizing the strengths of our experimental results, as well as our technical contributions, including the design of the Temporal Pyramid Representation (TPR), the integration of global and local feature extraction, and the spatiotemporal decoding.
>
> We address your specific concerns below and hope to clarify any misunderstandings.
>
> ### Q.1. Novelty of the Feature Extractors and Need for Additional Ablation Studies
> > 1. The feature extractors appear to be based on a Swin-Transformer and U-Net (Swin-Unet) combination, which raises concerns about novelty; additional ablation studies would help validate their effectiveness.
>
> Thank you for pointing this out. The novelty of our method arises from four key aspects:
> - Temporal Pyramid Representation (TPR) for capturing short-term dynamics: TPR allows our model to effectively capture motion information across multiple temporal scales, as shown in Figure 2 of the paper.
> - Modeling long-term motion with a multi-frame encoder: Unlike previous methods such as TimeLens, VideoINR, and MoTIF, which process only two frames, our encoder can handle multiple frames, enabling the capture of long-term dependencies (Ablation Studies Table, Cases 3, 6, and 7).
> - Dual-branch feature extraction for local and global information: We designed a feature extractor with separate branches to capture both local and global features during the encoding process, resulting in comprehensive spatiotemporal representations (Ablation Studies Table, Cases 1, 2, and 3).
> - Unified spatiotemporal decoder for simultaneous temporal and spatial upsampling: Our decoder integrates temporal and spatial dimensions cohesively, facilitating the reconstruction of frames at arbitrary times and resolutions (Ablation Studies Table, Cases 3, 4, and 5).
>
> We appreciate your suggestion regarding additional ablation studies. We will include further experiments in the revised version to validate the effectiveness of our feature extractors, thereby enhancing the robustness of our paper.
>
> ### Q.2. Inclusion of Video Metrics for Quality and Temporal Consistency
> > 2. Given that this is a space-time video super-resolution approach, including one or two video metrics to assess quality and temporal consistency would provide a more comprehensive evaluation.
>
> Thank you for this valuable suggestion. We agree that incorporating additional video metrics would strengthen our evaluation. In the revised version, we will include metrics to assess both visual quality and temporal consistency comprehensively.
>
> ### Q.3 Limited Visual Comparisons with Baseline Methods
> > 3. The visual comparison results include only one or two baseline methods, while more methods are compared in the tabular results.
> We appreciate your observation. We have updated the manuscript to include new visualization results that encompass a broader range of baseline methods. This addition provides a more comprehensive visual comparison and better illustrates the advantages of our approach.
>
> ### Q.4 Handling Longer Event Sequences Compared to Other Methods
> > 4. It appears that the proposed method can handle a longer sequence of events as input. In comparison, what is the length of the event input used by other methods in this study? Given that a long event sequence may not be suitable for other methods to provide optimal results, testing different sequence lengths would be beneficial.
>
> Thank you for your thoughtful question. Our method aligns the input events with the corresponding frame rates, effectively handling longer sequences. In the Ablation Studies Table (Cases 3, 6, and 7), we discuss the impact of varying the number of input frames on performance. It is important to emphasize that the ability to process multiple frames is a significant advantage of our method, as previous approaches were limited to two-frame inputs and could not fully exploit long-term temporal information.
>
> We hope that our responses address your concerns satisfactorily. Your constructive feedback has been instrumental in improving our work, and we are grateful for your contributions.
>
> Sincerely,

---

### Official Review · Reviewer_5cxJ · 2024-11-03

**Soundness:** 2
**Presentation:** 2
**Contribution:** 2
**Rating:** 3
**Confidence:** 5

**Summary:**

The paper addresses Continuous Space-Time Video Super-Resolution (C-STVSR), which aims to upscale both resolution and frame rate in videos at arbitrary scales. Existing methods that use implicit neural representation (INR) struggle with capturing rapid, nonlinear motion, and long-term dependencies due to their reliance on linear motion assumptions and limited frame usage. To address these limitations, the authors propose HR-INR, a C-STVSR framework enhanced by event  signals, which provide high temporal resolution. The framework includes a regional event feature extractor that captures nonlinear motion through an event temporal pyramid and a holistic event-frame feature extractor for long-term dependencies. Additionally, an INR-based decoder with spatiotemporal embeddings is introduced to handle long-term dependencies across a large temporal scope. Results on both synthetic and real-world datasets demonstrate the proposed method’s effectiveness and generalization capabilities.

**Strengths:**

1. In the STVSR task, the paper effectively addresses the limitations of RGB-based methods in handling highly dynamic motion by introducing event signals, which helps improve performance.
2. Compared to existing event representation methods, such as voxel grids, time surfaces, and time moments, this paper proposes an innovative event representation method, the Temporal Pyramid Representation (TPR).

**Weaknesses:**

1. The paper claims that compared to existing RGB-based methods such as VideoINR and MoTIF, the use of event signals can address highly dynamic motion, which is intuitive. However, the assertion that extending the input window to four frames solves the long-term dependency issue lacks explanation and experimental validation. For instance, it would be helpful to clarify the specific scenarios where a four-frame window is essential, and a two-frame setup would be insufficient. A suggestion would be to visualize Temporal Attention on the input images in Figure 4.
2. While the introduction of event signals may address highly dynamic motion, the paper does not discuss any potential challenges introduced by event signals. The scientific problem being addressed also remains unclear. This paper seems to be engineering-oriented without scientific rigor.
3. The paper claims its approach avoids the gaps and holes commonly associated with optical flow methods. However, in comparison with the optical flow-based VideoINR, no explanation or experimental evidence demonstrates how this approach mitigates these issues, or why VideoINR cannot mitigate them. For example, in Figure 13, the HR-INR method exhibits blurring in frame 000080 and noticeable holes in the hand region of frame 000160, indicating that gaps and holes persist in certain cases. As a result, the novelty of this work relative to previous studies is not established.

4. The contributions of the paper are not clearly defined. It is recommended that the authors refine and explicitly summarize the key contributions of the work in the writing.

Overall, I think
1. the proposed method lacks a clearly defined contribution to the STVSR task. It seems to be too engineering-oriented.
2. the experimental results are insufficient to demonstrate a clear advantage over RGB-based methods.

**Questions:**

1. Why does Figure 8 compare the proposed method with the sub-SOTA method, TimeLens, rather than the more recent SOTA method, such as CBMNet?

---

> ### Author Response · Authors · 2024-11-22
> **Response to Reviewer 5cxJ's Comments and Feedback (1/2)**
>
> # Response to Reviewer 5cxJ's Comments and Feedback (1/2)
>
> Dear Reviewer 5cxJ,
>
> Thank you for your insightful comments and for taking the time to review our work.
>
> ### Q.1 Need for Explanation and Experimental Validation of Long-Term Dependency Modeling
>
> > 1. The paper claims that compared to existing RGB-based methods such as VideoINR and MoTIF, the use of event signals can address highly dynamic motion, which is intuitive. However, the assertion that extending the input window to four frames solves the long-term dependency issue lacks explanation and experimental validation. For instance, it would be helpful to clarify the specific scenarios where a four-frame window is essential, and a two-frame setup would be insufficient. A suggestion would be to visualize Temporal Attention on the input images in Figure 4.
>
> We appreciate your thoughtful feedback. We have discussed the use of multi-frame inputs in our ablation studies, specifically in Table 5, Cases 3, 6, and 7. These experiments demonstrate the effectiveness of modeling long-term dependencies by using multiple frames.
>
> In response to your suggestion, we will include visualizations of the temporal attention in Figure 4 of the revised paper. This addition will help clarify the specific scenarios where a four-frame input is crucial compared to a two-frame setup.
>
>
> ### Q.2 Lack of Discussion on Challenges Introduced by Event Signals and Clarity of the Scientific Problem
>
> > 2. While the introduction of event signals may address highly dynamic motion, the paper does not discuss any potential challenges introduced by event signals. The scientific problem being addressed also remains unclear. This paper seems to be engineering-oriented without scientific rigor.
>
> Thank you for highlighting this important point.
> We define the scientific problem of this paper we are addressing in Line 75.
> Specifically, our research focuses on how to leverage event data to guide spatiotemporal upsampling in videos.
>
> We will also emphasize the two main technical challenges we tackle:
> - Capturing rapid motion: How to effectively model fast-moving objects using event data.
> - Modeling long-term motion dependencies: How to handle temporal dependencies over longer sequences.
>
> To address these challenges, we designed the Temporal Pyramid Representation (TPR) and a dual-branch encoder that combines local and global features. We have validated the effectiveness of these designs through our experiments.
>
> We appreciate your emphasis on scientific rigor, and we believe that clarifying these points will enhance the clarity and impact of our paper.
>
> ### Q.3 Lack of Evidence on Mitigating Gaps and Holes Compared to Optical Flow Methods
>
> > 3. The paper claims its approach avoids the gaps and holes commonly associated with optical flow methods. However, in comparison with the optical flow-based VideoINR, no explanation or experimental evidence demonstrates how this approach mitigates these issues, or why VideoINR cannot mitigate them. For example, in Figure 13, the HR-INR method exhibits blurring in frame 000080 and noticeable holes in the hand region of frame 000160, indicating that gaps and holes persist in certain cases. As a result, the novelty of this work relative to previous studies is not established.
>
> Thank you for your constructive criticism regarding the novelty and limitations of our method. In Figures 13 and 5, as well as in the supplementary videos, our method demonstrates fewer gaps and holes compared to optical flow-based methods.
>
> Artifacts such as gaps and holes are common challenges in interpolation tasks. Our visual examples illustrate that our approach produces fewer of these artifacts. We acknowledge that some blurring and holes may still occur in complex scenarios, and we will use more precise language in the paper to accurately describe the improvements our method offers over existing approaches.

---

> > ### Author Response · Authors · 2024-11-22
> > **Response to Reviewer 5cxJ's Comments and Feedback (2/2)**
> >
> > # Response to Reviewer 5cxJ's Comments and Feedback (2/2)
> >
> > ### Q.4. Contributions of the Paper Not Clearly Defined
> >
> > > 4. The contributions of the paper are not clearly defined. It is recommended that the authors refine and explicitly summarize the key contributions of the work in the writing.
> >
> > We appreciate your suggestion and agree that clearly defining our contributions will strengthen the paper. In the revised version, we will explicitly summarize the key contributions in both the introduction and conclusion sections.
> >
> > Our main contributions are:
> >
> > 1. Novel problem formulation: Investigating how event data can simultaneously guide spatial and temporal video enhancement.
> > 2. Technical innovations:
> >     - (I) Proposing the Event Temporal Pyramid Representation (TPR) to capture short-term dynamic motion.
> >     - (II) Designing a feature extraction process that combines holistic and regional features to manage motion dependencies.
> >     - (III) Developing a spatiotemporal decoding process based on implicit neural representation, which avoids traditional optical flow methods and achieves stable frame interpolation through temporal-spatial embedding.
> >
> > ### Q.5. Method Appears Too Engineering-Oriented Without Clear Contribution to STVSR Task
> >
> > > 5. the proposed method lacks a clearly defined contribution to the STVSR task. It seems to be too engineering-oriented.
> >
> > Thank you for your perspective. Our method addresses key challenges in the STVSR task, specifically capturing rapid motion and modeling long-term dependencies. We have made technical contributions to tackle these challenges, as outlined above. Additionally, our method demonstrates improved visual results in practical applications. We will enhance the writing in the revised paper to better highlight these contributions and their significance to the STVSR task.
> >
> > ### Q.6 Experimental Results Insufficient to Demonstrate Clear Advantage Over RGB-Based Methods
> >
> > > 6. the experimental results are insufficient to demonstrate a clear advantage over RGB-based methods.
> >
> > We value your feedback. Our method shows significant advantages over RGB-based methods. For example, in Figure 5, our approach successfully reconstructs the motion of spinning wheels, which RGB-based methods fail to capture. We have also included additional visual results in the supplementary material. We encourage you to review these examples, which further demonstrate the effectiveness of our method.
> >
> > ### Q.7. Comparison with More Recent SOTA Methods Like CBMNet
> >
> > > 7. Why does Figure 8 compare the proposed method with the sub-SOTA method, TimeLens, rather than the more recent SOTA method, such as CBMNet?
> >
> > Thank you for bringing this to our attention. In the revised paper, we have added visual comparisons with more recent state-of-the-art methods, including CBMNet. This inclusion provides a more comprehensive evaluation of our method's performance relative to the latest advances in the field.
> >
> > We hope that our responses address your concerns satisfactorily. Your constructive feedback has been instrumental in improving our work, and we are grateful for your thorough review.

---

### Official Review · Reviewer_KpVS · 2024-11-04

**Soundness:** 3
**Presentation:** 3
**Contribution:** 2
**Rating:** 6
**Confidence:** 4

**Summary:**

Summary:

This paper addresses the challenging task of Continuous Space-Time Video Super-Resolution (C-STVSR), which aims to enhance video resolution and frame rate at arbitrary scales. Experiments demonstrate HR-INR’s effectiveness across four datasets, achieving state-of-the-art results.

**Strengths:**

Strengths

1.The authors’ novel approach of integrating an event camera for high-resolution temporal information is commendable. Event cameras capture finer temporal details than traditional cameras, making them particularly effective for dynamic scenes with nonlinear motion. This approach allows the method to move beyond traditional frame-based video super-resolution limitations.

2.The model is tested on a variety of datasets, both simulated and real, highlighting its robustness and generalization ability. The results indicate HR-INR’s superiority over prior methods, even achieving better performance with much fewer parameters than other models.

**Weaknesses:**

Weaknesses:

1.It appears that this paper achieves better performance and accomplishes more tasks with fewer parameters (e.g., one-tenth of TimeLens’s parameters). Can the authors analyze whether their method has surpassed existing methods in both super-resolution and interpolation tasks? Are there any limitations?

2.In the experimental setup, the authors used Adobe24 + Vid2e to generate events for pretraining and tested on other datasets. If I’m not mistaken, the training data for other methods, aside from the CED dataset, is different from that used in this paper. Although the authors emphasize that TimeLens used a larger training set, it seems that event density, which affects the performance of event-based tasks, was not standardized across methods.

3.For the analysis of the Temporal Representation Pyramid (TRP), is there actually motion within 1/1000s in the current datasets?

**Questions:**

Questions:

Considering that this paper does not commit to open-source code, do the authors have any plans to release the code?

---

> ### Author Response · Authors · 2024-11-21
> **Response to Reviewer KpVS's Comments and Feedback**
>
> Dear Reviewer KpVS,
>
> Thank you for recognizing the novelty of our method and its strong performance across multiple datasets. Your positive feedback greatly encourages us.
>
> ### Q.1. Analysis of performance in super-resolution and interpolation tasks, and limitations
> > It appears that this paper achieves better performance and accomplishes more tasks with fewer parameters (e.g., one-tenth of TimeLens’s parameters). Can the authors analyze whether their method has surpassed existing methods in both super-resolution and interpolation tasks? Are there any limitations?
> Thank you for your insightful question regarding whether our method surpasses existing approaches in both super-resolution and interpolation tasks, and any associated limitations. The advantages of our method over previous work stem from four main aspects:
>
> 1. **Fine-grained capture of local (regional) motion:** We designed the Event Temporal Pyramid Representation (TPR) to capture motion information over both long and short durations, as illustrated in Figure 2 of the main text. The effectiveness of TPR is demonstrated in our ablation studies (Tables 5 and 6).
> 2. **Modeling longer (holistic) temporal sequences (e.g., four frames):** In contrast to prior methods like TimeLens, our approach can model motion across multiple frames. This capability is validated in our ablation studies (Table 5, Cases 3, 6, and 7).
> 3. **Joint local and global feature extraction:** Building upon our local motion capture and long-term inputs, we designed a feature extractor that integrates both global and local features. This extractor effectively captures information at different scales, as discussed in our ablation studies (Table 5, Cases 1, 2, and 3).
> 4. **Unified spatiotemporal INR decoder:** We introduced a unified Implicit Neural Representation (INR) decoder capable of reconstructing frames at arbitrary times and resolutions. Analytical experiments regarding this decoder are presented in Table 5, Cases 3, 4, and 5.
>
> Overall, these four aspects contribute to the advancements of our method over prior work. However, our method also has certain limitations, such as extended training times and potential color distortions, which we have discussed in the Base Case analysis.
>
> ### Q.2. Differences in training data and event density standardization
> > 2. In the experimental setup, the authors used Adobe24 + Vid2e to generate events for pretraining and tested on other datasets. If I’m not mistaken, the training data for other methods, aside from the CED dataset, is different from that used in this paper. Although the authors emphasize that TimeLens used a larger training set, it seems that event density, which affects the performance of event-based tasks, was not standardized across methods.
>
> We appreciate your astute observation about the differences in training data and the standardization of event density across methods. Indeed, as you pointed out, this reflects the current state of the field. Currently, real event-based interpolation datasets are relatively small, which is insufficient for fully training interpolation networks to convergence. Consequently, similar to previous methods such as TimeLens, we first pre-trained our model on simulated datasets (Adobe24 + Vid2e) and then fine-tuned it on real datasets. This approach helps to mitigate issues related to varying event densities across different datasets.
>
> ### Q.3. Motion within 1/1000th of a second in current datasets for TRP analysis
> > 3. For the analysis of the Temporal Representation Pyramid (TRP), is there actually motion within 1/1000s in the current datasets?
>
> Thank you for your meticulous observation regarding the Temporal Representation Pyramid (TRP). The TRP is indeed capable of modeling motion occurring within 1/1000th of a second. In the BS-ERGB dataset, we utilized the Prophesee event camera sensor, which possesses a temporal resolution equivalent to 10,000 fps (i.e., 1/10,000th of a second). This dataset includes numerous high-speed motion scenes where motion within 1/1000th of a second is significant. An example of such motion is the basketball scene depicted in Figure 10. We will include additional examples to further illustrate this point.
>
> ### Q.4 Plans for code release
> > 4. Considering that this paper does not commit to open-source code, do the authors have any plans to release the code?
>
> We are committed to open-sourcing all our code and pre-trained models once the paper is accepted for publication.
>
> Thank you again for your valuable feedback. We look forward to further discussions with you, which will be of great help in improving our work.

---

> > ### Comment · Reviewer_KpVS · 2024-11-26
> >
> > Thank you for your response. I believe the work presented in this paper is thoroughly well-executed, and I will retain my current score.

---

### Author Response · Authors · 2024-12-03
**Summary of Discussion and Revision Updates (1/2)**

Dear Reviewers,

We sincerely thank you for your valuable feedback and constructive comments. Based on your insights, we have revised the paper to address your concerns and further strengthen our contributions. As the discussion phase approaches its conclusion, we would like to restate the contributions of our work and summarize the updates made in the revision.

### Contributions

1. **Novel Problem Formulation:**
   - This work is the first to address the novel scientific problem of using events to simultaneously enhance the spatial and temporal resolution of videos, achieving state-of-the-art performance. [KpVS.Strength.1, 5cxJ.Strength.1, fyeD.Strength.1]
   - Compared to frame-based methods [1,2], our approach leverages events to compensate for inter-frame information, enabling the modeling of complex nonlinear motion and capturing rapid inter-frame motion more precisely. [5cxJ.Strength.2]
   - Compared to event-based methods [3,4,5], our method captures long-term motion dependencies while enhancing both spatial and temporal information. [5cxJ.Strength.1]
   - The proposed method demonstrates superior performance across five datasets.

2. **Technical Innovations:**
   - This paper proposes three key technical contributions to address challenges in capturing rapid motion, modeling long-term dependencies ($>= 4$ frames), and decoding spatiotemporal information. [KpVS.Strength.1, wQx2.Strength.1, fyeD.Strength.2]
     1. A **Temporal Pyramid Representation (TPR)** for events, which effectively captures millisecond-level motion. [5cxJ.Strength.2, wQx2.Strength.1]
     2. A Regional-Holistic feature extraction framework to model both short-term motion and long-term dependencies. [wQx2.Strength.2]
     3. An **INR-based spatiotemporal decoder** that produces stable interpolation and super-resolution results. [wQx2.Strength.3, fyeD.Strength.2]

3. **Comprehensive Experiments and Analyses:**
   - Comparative experiments on five datasets validate the robustness and effectiveness of our approach. [KpVS.Strength.2, fyeD.Strength.1]
     - Tables 1, 2, 3, 4, and 9 provide quantitative comparisons.
   - Analytical experiments further highlight the contributions of each component.
     - Tables 5, 6, 7, and 8, as well as Sections 4.2 and D, provide detailed analyses.
   - Extensive visualizations support these findings (Figures 5, 6, and 7–15).

In summary, this work pioneers the problem of event-guided spatiotemporal video super-resolution, proposes effective solutions, and validates their effectiveness through extensive experiments.

---

### Author Response · Authors · 2024-12-03
**Summary of Discussion and Revision Updates (2/2)**

### Updates in the Revision

We deeply appreciate the reviewers' insightful comments, which have helped us improve the paper. While most of the comments have been addressed in the official comments, we provide additional experimental results and analyses in the revision. The updates include:

1. **Comparison with Cascade Pipelines:**
   - Added new comparisons with cascade pipelines in **Tables 1 and 2 (Page 8)**, providing stronger evidence of our method's robustness. [fyeD.Q.4]

2. **Event Generation and its Role in C-STVSR:**
   - **Section A (Pages 16–17)** explains the principles of event generation and their benefits for spatiotemporal super-resolution. [5cxJ.Q.2]

3. **Comparison with CBMNet in Real-World Scenarios:**
   - **Section D.1 (Pages 19–21)** includes new comparisons with CBMNet, demonstrating superior interpolation performance across various object scales and motion amplitudes. [5cxJ.Q.7, wQx2.Q.3]

4. **Effectiveness of Temporal Pyramid Representation (TPR):**
   - **Section D.2 (Pages 22–23)** provides analyses showing TPR’s ability to capture both global and local motion effectively. [KpVS.Q.3]

5. **Advantages of Multi-Frame Inputs:**
   - **Section D.3 (Pages 22, 24)** demonstrates how multi-frame inputs enhance long-term motion modeling through feature visualizations and improved interpolation results. [5cxJ.Q.1, wQx2.Q.4]

6. **No-Reference Metric Evaluation:**
   - **Section D.4 (Page 24)** includes new results on video interpolation evaluated using no-reference metrics. [wQx2.Q.2]

7. **Experiments on the APLIX-VSR Dataset:**
   - **Section D.5 (Pages 24–25)** presents additional experiments on the APLIX-VSR dataset, further validating our method's generalizability. [fyeD.Q.5]

Thank you again for your constructive feedback, which has greatly improved the quality of our work.

Sincerely,


### Reference:
- [1] Chen Song, Qixing Huang, and Chandrajit Bajaj. E-cir: Event-enhanced continuous intensity recovery. In Proceedings of the IEEE/CVF Conference on Computer Vision and Pattern Recognition, pp. 7803–7812, 2022.
- [2] Yi-Hsin Chen, Si-Cun Chen, Yen-Yu Lin, and Wen-Hsiao Peng. Motif: Learning motion trajectories with local implicit neural functions for continuous space-time video super-resolution. In Proceedings of the IEEE/CVF International Conference on Computer Vision, pp. 23131–23141, 2023b.
- [3] Stepan Tulyakov, Daniel Gehrig, Stamatios Georgoulis, Julius Erbach, Mathias Gehrig, Yuanyou Li, and Davide Scaramuzza. Time lens: Event-based video frame interpolation. In Proceedings of the IEEE/CVF Conference on Computer Vision and Pattern Recognition, pp. 16155–16164, 2021.
- [4] Stepan Tulyakov, Alfredo Bochicchio, Daniel Gehrig, Stamatios Georgoulis, Yuanyou Li, and Davide Scaramuzza. Time lens++: Event-based frame interpolation with parametric non-linear flow and multi-scale fusion. In Proceedings of the IEEE/CVF Conference on Computer Vision and Pattern Recognition, pp. 17755–17764, 2022.
- [5] Taewoo Kim, Yujeong Chae, Hyun-Kurl Jang, and Kuk-Jin Yoon. Event-based video frame interpolation with cross-modal asymmetric bidirectional motion fields. In Proceedings of the IEEE/CVF Conference on Computer Vision and Pattern Recognition, pp. 18032–18042, 2023.

---

### Meta-Review · Area_Chair_TcMP · 2024-12-21

**Metareview:**

## Summary

The paper presents a novel Continuous Space-Time Video Super-Resolution (C-STVSR) framework, HR-INR, which enhances video resolution and frame rate at arbitrary scales. The framework uses event signals, event temporal pyramid representation, event-frame feature extractor, and spatiotemporal decoding to capture both holistic dependencies and regional motions, outperforming existing techniques.

## Strength

* Integrates an event camera for high-resolution temporal information, enhancing its effectiveness for dynamic scenes with nonlinear motion.
* Tests on various datasets, demonstrating its robustness and generalization ability.
* Proposes Temporal Pyramid Representation (TPR) as an innovative event representation method and enhances the method's temporal and spatial consistency with the spatial-temporal decoding module.
* Achieves state-of-the-art performance on both VSR and VFI using events.

## Weaknesses

* The experimental setup used in the study is different from other methods, suggesting that event density, which affects event-based tasks, is not standardized across methods.
* The Temporal Representation Pyramid (TRP) analysis is also unclear, with no clear explanation or experimental validation for the use of event signals.
* The paper's approach to optical flow methods is not well-defined, with gaps and holes persisting in certain cases, indicating the novelty of the work.
* The paper's technical aspects include unclear definitions of the Inverse Neural Network (INR) and the use of regional and holistic events.

## Conclusion
First of all, thanks to the authors for the enormous effort to provide new results and comparative analysis to the paper and try to answer all the doubts of the reviewers. However, all the new content should be summarized in the main manuscript and have a second review including the improvements suggested by the reviewers.`

**Additional Comments On Reviewer Discussion:**

First of all, thanks to the authors for the enormous effort to provide new results and comparative analysis to the paper and try to answer all the doubts of the reviewers. However, all the new content should be summarised in the main manuscript and have a second review including the improvements suggested by the reviewers.`

---

### Decision · Program_Chairs · 2025-01-22

Reject